# LRMP inhibits cAMP potentiation of HCN4 channels by disrupting intramolecular signal transduction

Colin H Peters[1†], Rohit K Singh[1,2†], Avery A Langley[1], William G Nichols[1], Hannah R Ferris[1], Danielle A Jeffrey[1], Catherine Proenza[1,3*], John R Bankston[1*]

[1]Department of Physiology and Biophysics, University of Colorado Anschutz Medical Campus, Aurora, United States; [2]Skaggs School of Pharmacy, Department of Pharmaceutical Sciences, University of Colorado Anschutz Medical Campus, Aurora, United States; [3]Department of Medicine, Division of Cardiology, University of Colorado Anschutz Medical Campus, Aurora, United States

*For correspondence:
catherine.proenza@cuanschutz.
edu (CP);
john.bankston@cuanschutz.edu
(JRB)

†These authors contributed
equally to this work

Competing interest: The authors
declare that no competing
interests exist.

Reviewing Editor: Henry M
Colecraft, Columbia University,
United States

**Abstract** Lymphoid restricted membrane protein (LRMP) is a specific regulator of the hyperpolarization-activated cyclic nucleotide-sensitive isoform 4 (HCN4) channel. LRMP prevents cAMP-dependent potentiation of HCN4, but the interaction domains, mechanisms of action, and basis for isoform-specificity remain unknown. Here, we identify the domains of LRMP essential for this regulation, show that LRMP acts by disrupting the intramolecular signal transduction between cyclic nucleotide binding and gating, and demonstrate that multiple unique regions in HCN4 are required for LRMP isoform-specificity. Using patch clamp electrophysiology and Förster resonance energy transfer (FRET), we identified the initial 227 residues of LRMP and the N-terminus of HCN4 as necessary for LRMP to associate with HCN4. We found that the HCN4 N-terminus and HCN4-specific residues in the C-linker are necessary for regulation of HCN4 by LRMP. Finally, we demonstrated that LRMP-regulation can be conferred to HCN2 by addition of the HCN4 N-terminus along with mutation of five residues in the S5 region and C-linker to the cognate HCN4 residues. Taken together, these results suggest that LRMP inhibits HCN4 through an isoform-specific interaction involving the N-terminals of both proteins that prevents the transduction of cAMP binding into a change in channel gating, most likely via an HCN4-specific orientation of the N-terminus, C-linker, and S4-S5 linker.

## eLife assessment

This study identifies the molecular determinants of LRMP co-regulation of HCN 4 activity. The evidence supporting the conclusions, which is **compelling**, is backed by rigorous electrophysiological and spectroscopic analysis. The work is **important** because it greatly enhances our understanding of the mechanisms of HCN channel regulation in a tissue-specific manner and highlights a functional role for more disordered regions that have yet to be structurally resolved.

## Introduction

Hyperpolarization-activated, cyclic nucleotide-sensitive (HCN) ion channels are biophysical anomalies. Despite being structurally related to voltage-gated K$^+$ channels — which are activated by membrane depolarization and are highly selective for K$^+$ over Na$^+$ — HCN channels activate in response to membrane hyperpolarization and pass a mixed Na$^+$/K$^+$ current. In addition, binding of cyclic nucleotides, particularly cAMP, to a conserved C-terminal cyclic nucleotide binding domain (CNBD)

potentiates HCN channels by shifting the voltage-dependence of activation to more depolarized potentials, speeding activation, and slowing deactivation (*DiFrancesco and Tortora, 1991*; *Wainger et al., 2001*).

While the details of intramolecular transduction between cAMP binding and channel gating have yet to be fully elucidated, some key aspects are known: (1) the unbound CNBD is inhibitory — truncation of the CNBD potentiates channel activation, similar to cAMP binding to the intact CNBD (*Wainger et al., 2001*); (2) the slowing of channel deactivation in response to cAMP binding occurs through a separate mechanism from the shift in activation voltage dependence, and cannot be replicated by truncation of the CNBD (*Wicks et al., 2011*; *Sunkara et al., 2018*); and (3) transduction of the signal for the cAMP-dependent shift in channel activation, but not deactivation, requires interactions of a 'a cAMP transduction centre' (*Porro et al., 2019*; *Wang et al., 2020a*) comprised of portions of the C-linker (which connects the CNBD to the transmembrane domain), the N-terminal HCN domain (HCND), and the S4-S5 linker.

The inositol 1,4,5-triphosphate receptor-associated proteins, IRAG1 and LRMP/IRAG2 (lymphoid restricted membrane protein), are a family of endoplasmic reticulum (ER) transmembrane proteins that are isoform-specific regulators of HCN4 (*Peters et al., 2020*; *Peters et al., 2022*). IRAG1 and LRMP act by modulating the cAMP sensitivity of HCN4. However, they have opposing effects: IRAG1 causes a gain-of-function by shifting HCN4 activation to more depolarized membrane potentials in the absence of cAMP. In contrast, LRMP causes a loss-of-function by inhibiting cAMP-dependent potentiation of HCN4 activation (*Peters et al., 2020*). IRAG1 and LRMP share some sequence homology, particularly in their coiled-coil motifs, and both have been found to regulate IP$_3$ receptor calcium release channels (*Schlossmann et al., 2000*; *Geiselhöringer et al., 2004*; *Prüschenk et al., 2021*). However, the interaction domains, mechanisms of action, and basis for isoform-specificity for their regulation of HCN4 remain unknown.

In this study, we focused on LRMP and investigated the interaction domains and mechanism by which it inhibits the cAMP-dependent shift in HCN4 activation. Our previous study showed that LRMP acts differently from TRIP8b, a neuronal protein that also prevents the cAMP-dependent shift in HCN channel activation. TRIP8b acts by directly antagonizing cAMP binding (*Santoro et al., 2004*; *Zolles et al., 2009*; *Bankston et al., 2017*; *Saponaro et al., 2018*). In contrast, LRMP doesn't inhibit cAMP binding to the CNBD, as indicated by the preserved cAMP-dependent slowing of deactivation in the presence of LRMP (*Peters et al., 2020*). Furthermore, TRIP8b regulates all HCN channel isoforms (*Zolles et al., 2009*; *Santoro et al., 2011*), whereas LRMP is specific for the HCN4 isoform (*Peters et al., 2020*). These observations suggest that LRMP regulates HCN4 by interfering with an isoform-specific step in the signal transduction pathway that links cAMP binding to the shift in activation voltage-dependence.

We tested this hypothesis using a combination of patch clamp electrophysiology and FRET. We found that the initial N-terminal 227 residues of LRMP associate with the N-terminus of HCN4 and that the intact HCN4 N-terminus is required for channel regulation by LRMP. Furthermore, we show that two HCN4-specific residues in the C-linker, P545 and T547, are critical for isoform-specific regulation of HCN4 by LRMP. Finally, we found that addition of the HCN4 N-terminus along with mutations in the C-linker and S5 transmembrane segment are sufficient to confer LRMP regulation to HCN2. These results are consistent with a model in which LRMP inhibits HCN4 via the cAMP transduction centre (*Weißgraeber et al., 2017*; *Porro et al., 2019*; *Wang et al., 2020a*; *Saponaro et al., 2021*; *Kondapuram et al., 2022*).

## Results

### The N-terminus of LRMP is necessary and sufficient to regulate HCN4

We previously showed that LRMP significantly reduces the cAMP-dependent depolarizing shift in HCN4 activation without any effect on the voltage dependence in the absence of cAMP, and that it does not regulate HCN1 or HCN2 (*Peters et al., 2020*). We next sought to identify a subdomain within LRMP that is responsible for this regulation. We began with a truncated LRMP construct with a Citrine fluorescent protein replacing the C-terminal ER transmembrane and lumenal domains (LRMP 1-479Cit; *Figure 1A–D*; *Table 1*). We found that LRMP 1-479Cit inhibited the cAMP sensitivity of HCN4 to an even greater degree than the full-length LRMP (*Table 2*). This more pronounced effect

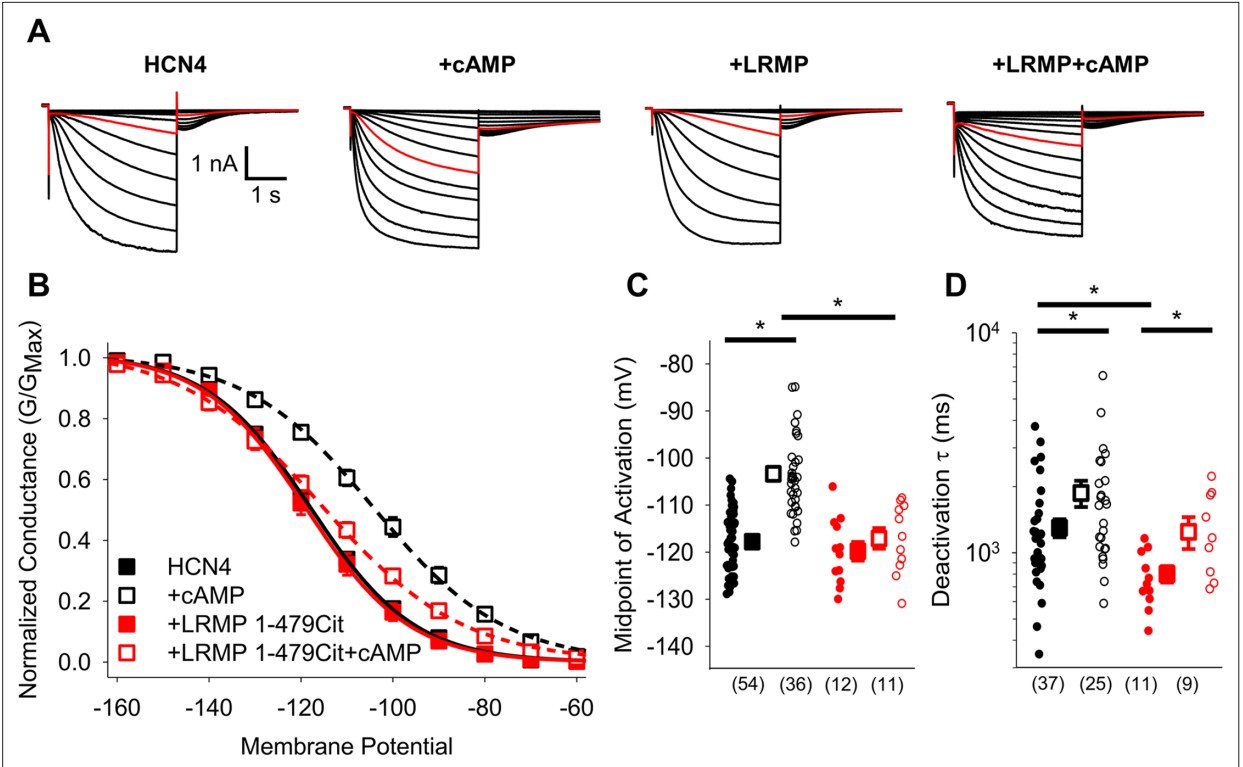

**Figure 1.** The cytosolic region of LRMP regulates HCN4 but does not antagonize cAMP binding. (**A**) Exemplar current recordings from HCN4 in the absence or presence of 1 mM cAMP and/or LRMP 1-479Cit. Currents recorded with a –110 mV activating pulse are shown in *red*. (**B**) Voltage dependence of activation for HCN4 alone (*black*) or co-transfected with LRMP 1-479Cit (*red*) in the presence or absence of 1 mM intracellular cAMP (*open symbols*). (**C**) Average (± standard error of the mean) midpoints of activation for HCN4 in the absence or presence of LRMP 1-479Cit and/or 1 mM cAMP using the same color scheme as (**B**). (**D**) Average (± standard error of the mean) time constants of deactivation for HCN4 in the absence or presence of LRMP 1-479Cit and/or 1 mM cAMP using the same color scheme as (**B**). Small circles in (**C**) and (**D**) represent individual cells and values in parentheses are the number of independent recordings for each condition. * indicates a significant (p<0.05) difference. All means, standard errors, and exact p-values are in *Table 1*.

**Table 1.** Midpoints of activation in HCN4 in the presence of LRMP fragments.

|  | Control (mV) | cAMP (1 mM) (mV) | ΔV½ in cAMP | p-Value Control vs. cAMP |
|---|---|---|---|---|
| HCN4 Control | –117.8±0.9 (54) | –103.4±1.5 (36) | 14.4 mV | *p*<0.0001 |
| HCN4 LRMP 1-479Cit | –119.8±2.0 (12) p=0.3847 | –117.1±2.2 (11) p<0.0001 | 2.7 mV | p0.3710 |
| HCN4 LRMP 1–227 | –117.9±1.9 (13) p=0.9399 | –118.1±1.4 (11) p<0.0001 | –0.2 mV | p=0.9642 |
| HCN4 LRMP 228–539 | –116.1±2.6 (9) p=0.5265 | –106.3±2.0 (8) p=0.3100 | 9.8 mV | p=0.0069 |
| HCN4 LRMP 1-108Cit | –123.1±1.8 (7) p=0.0747 | –103.0±2.5 (8) p=0.8890 | 20.1 mV | p<0.0001 |
| HCN4 LRMP 110-230Cit | –118.0±4.0 (9) p=0.9423 | –106.4±1.3 (12) p=0.2118 | 11.6 mV | p=0.0005 |

Average midpoint of activation (mV) ± standard error of the mean (Number of independent cells). ΔV½ values reflect the difference in population midpoints for whole-cell experiments in the presence vs. absence of cAMP.

**Table 2.** Midpoints of Activation for HCN Channel Constructs.

| | Control (mV) | +LRMP (mV) | LRMP vs. Control | Control ΔV½ in cAMP (mV) | LRMP ΔV½ in cAMP (mV) |
|---|---|---|---|---|---|
| HCN4 +cAMP | −117.8±0.9 (54)* −103.4±1.5 (36) p<0.0001 | −120.1±2.2 (14)* −114.7±2.6 (16) p=0.0724 | p=0.3530 **p<0.0001** | 14.4 mV | 5.4 mV |
| HCN2 +cAMP | −109.3±1.5 (8) −90.3±3.2 (8) p<0.0001 | −114.4±1.9 (8) −87.9±1.6 (8) p<0.0001 | p=0.1101 p=0.4293 | 19.0 mV | 26.5 mV |
| HCN4 Δ1–25 +cAMP | −118.1±2.2 (19) −101.1±2.6 (13) p<0.0001 | −121.0±2.7 (10) −116.5±1.5 (12) p=0.2236 | p=0.3859 **p<0.0001** | 17.0 mV | 4.5 mV |
| HCN4 Δ1–62 +cAMP | −116.5±1.7 (8) −99.1±2.2 (10) p<0.0001 | −118.8±1.9 (10) −107.9±1.4 (8) p=0.0003 | p=0.4089 **p=0.0027** | 17.4 mV | 10.9 mV |
| HCN4 Δ1–130 +cAMP | −115.2±2.2 (11) −101.3±2.3 (8) p=0.0003 | −117.4±1.3 (6) −106.9±3.7 (7) p=0.0152 | p=0.5651 p=0.1481 | 13.9 mV | 10.5 mV |
| HCN4 Δ1–185 +cAMP | −117.1±2.1 (12) −103.1±3.3 (13) p<0.0001 | −125.5±2.3 (8) −103.6±3.1 (8) p<0.0001 | p=0.0500 p=0.8913 | 14.0 mV | 21.9 mV |
| HCN4 V604X +cAMP | −101.7±2.0 (12) −104.7±4.9 (6) p=0.4698 | −102.0±1.9 (6) — | p=0.9407 — | −3.0 mV | — |
| HCN4 S719X +cAMP | −124.9±1.3 (17) −106.5±1.6 (22) p<0.0001 | −121.4±1.6 (20) −114.1±1.6 (18) **p=0.0018** | p=0.1217 **p=0.0010** | 18.4 mV | 7.3 mV |
| HCN4 PT/AF +cAMP | −127.6±1.6 (21) −117.1±2.2 (15) p=0.0002 | −127.8±1.9 (15) −112.1±2.1 (15) p<0.0001 | p=0.9311 p=0.0785 | 10.5 mV | 15.7 mV |

*Table 2 continued on next page*

*Table 2 continued*

| | Control (mV) | +LRMP (mV) | LRMP vs. Control | Control ΔV½ in cAMP (mV) | LRMP ΔV½ in cAMP (mV) |
|---|---|---|---|---|---|
| HCN2 AF/PT +cAMP | −106.5±1.9 (15) −88.2±0.7 (16) o<0.0001 | −107.7±1.5 (11) −86.4±1.6 (13) p<0.0001 | p=0.5858 p=0.3987 | 18.3 mV | 21.3 mV |
| HCN4-2 +cAMP | −112.7±2.8 (11) −94.8±3.1 (15) p<0.0001 | −111.9±2.2 (14) −102.5±1.9 (16) p=0.0092 | p=0.8290 **p=0.0284** | 17.9 mV | 9.4 mV |
| HCN2 VVGPT +cAMP | −103.4±2.1 (11) −88.9±2.2 (10) p=0.0002 | −105.6±2.8 (10) −93.3±3.0 (9) p=0.0018 | p=0.5305 p=0.2278 | 14.5 mV | 12.3 mV |
| HCN2-4N VVGPT +cAMP | −104.6±2.1 (16) −89.6±2.3 (16) p<0.0001 | −100.9±2.3 (14) −99.9±1.2 (11) p=0.7574 | p=0.2183 **p=0.0019** | 15.0 mV | 1.0 mV |

Average midpoint of activation ± standard error of the mean (Number of independent cells). ΔV½ values reflect the difference in population midpoints for whole-cell experiments in the presence vs. absence of cAMP.

*HCN4 control and cAMP data in the absence of LRMP is the same as in *Table 1*.

may be due to improved expression of this tagged construct compared to the untagged full-length LRMP, which was detected by co-transfection with GFP, or it may be that removal of the ER-transmembrane segment increased the proximity between LRMP and HCN4 by allowing LRMP to diffuse freely in the cytosol. A key feature of LRMP in our original study is that it does not prevent binding of cAMP to the CNBD of HCN4. This was also the case for LRMP 1-479Cit, as indicated by the significant slowing of deactivation by cAMP even in the presence of LRMP 1-479Cit (p=0.0310; *Figure 1D*). These results indicate that the ER transmembrane and luminal domains of LRMP are not required for regulation of HCN4 and they support the idea that LRMP limits cAMP potentiation of HCN4 by interfering with a downstream step in the cAMP signal transduction pathway.

To further resolve which regions of LRMP are required to regulate HCN4, we tested a series of additional truncated LRMP constructs (shown schematically in *Figure 2A*, *Figure 2—figure supplement 1*) for their ability to prevent cAMP-dependent shifts in HCN4 activation. We first split LRMP into two fragments: the LRMP 1–227 construct contains the N-terminus of LRMP with a cut-site near the N-terminus of the predicted coiled-coil sequence, while LRMP 228–539 contains the remainder of the protein. We found that LRMP 1–227 recapitulated the effects of full-length LRMP, while LRMP 228–539 had no effect on HCN4 gating (*Figure 2B, C and F*; *Table 1*). However, when we further split the N-terminal domain of LRMP into two fragments tagged with C-terminal Citrines, neither LRMP 1-108Cit nor LRMP 110-230Cit regulated HCN4 (*Figure 2D–F*; *Table 1*). Thus, the first 227 residues of LRMP are sufficient to regulate HCN4 and it seems likely that residues in both halves of the LRMP N-terminus participate in this regulation.

## The N-terminus of HCN4 is required for regulation by LRMP

We next examined the domains of the HCN4 channel that are necessary for regulation by LRMP. Since LRMP regulates only the HCN4 isoform, we focused on the large non-conserved regions in the distal N- and C-terminals as potential sites for LRMP regulation. We first examined the N-terminus by testing the ability of LRMP to regulate a series of HCN4 channels with progressively larger truncations (Δ1–25, Δ1–62, Δ1–130, Δ1–185, and Δ1–200; shown schematically in *Figure 3A*). The four smaller deletions all produced functional channels with normal cAMP-dependent shifts in activation, albeit with smaller current amplitudes in HCN4 Δ1–130 and Δ1–185 (*Figure 3D and E insets*). The HCN4 Δ1–200 construct produced insufficient current amplitude for analysis.

When the first 25 residues in the HCN4 N-terminus were truncated, LRMP still prevented cAMP from shifting HCN4 activation, just as in the WT HCN4 channel (*Figure 3B and F*; *Table 2*). Truncation of residues 1–62 led to a partial LRMP effect where cAMP caused a significant depolarizing shift in the presence of LRMP, but the activation in the presence of LRMP and cAMP was hyperpolarized compared to cAMP alone (*Figure 3C and F*; *Table 2*). In the HCN4 Δ1–130 construct, cAMP caused a significant depolarizing shift in the presence of LRMP; however, the midpoint of activation in the presence of LRMP and cAMP showed a non-significant trend towards hyperpolarization compared to cAMP alone (*Figure 3D and F*; *Table 2*). Finally, truncation of the first 185 residues, which removes most of the non-conserved region of the HCN4 N-terminus, completely abolished LRMP regulation of the channel (*Figure 3E and F*; *Table 2*); when LRMP was present, cAMP caused a significant depolarizing shift in the HCN4 Δ1–185 activation, and the midpoint of activation in the presence of both LRMP and cAMP was not significantly different from the midpoint in the presence of cAMP alone. These results suggest that the multiple subdomains within the non-conserved N-terminus of HCN4 are necessary for functional regulation by LRMP.

We also investigated LRMP regulation of two C-terminal truncations in HCN4: HCN4 S719X, which removes the C-terminus distal to the CNBD (*Liao et al., 2012*), and HCN4 V604X, which additionally removes the CNBD (shown schematically in *Figure 4A*). We found that truncation of the distal C-terminus (HCN4-S719X) reduced but did not eliminate LRMP regulation of HCN4. In the presence of both LRMP and cAMP, the activation of HCN4-S719X was still significantly hyperpolarized compared to the presence of cAMP alone (*Figure 4B and C*; *Table 2*). While cAMP caused a significant (~7 mV) shift in HCN4-S719X activation in the presence of LRMP, this was less than half the shift in the absence of LRMP (~18 mV). HCN4-V604X, which truncates the channel between the C-linker and CNBD, shifts channel activation to more depolarized potentials and completely prevents cAMP-dependent regulation (*Figure 4D and E*; *Table 2*), similar to the effects of the homologous HCN2-V526X mutant (*Wainger et al., 2001*). LRMP did not alter the gating of HCN4-V604X in the absence of cAMP, and

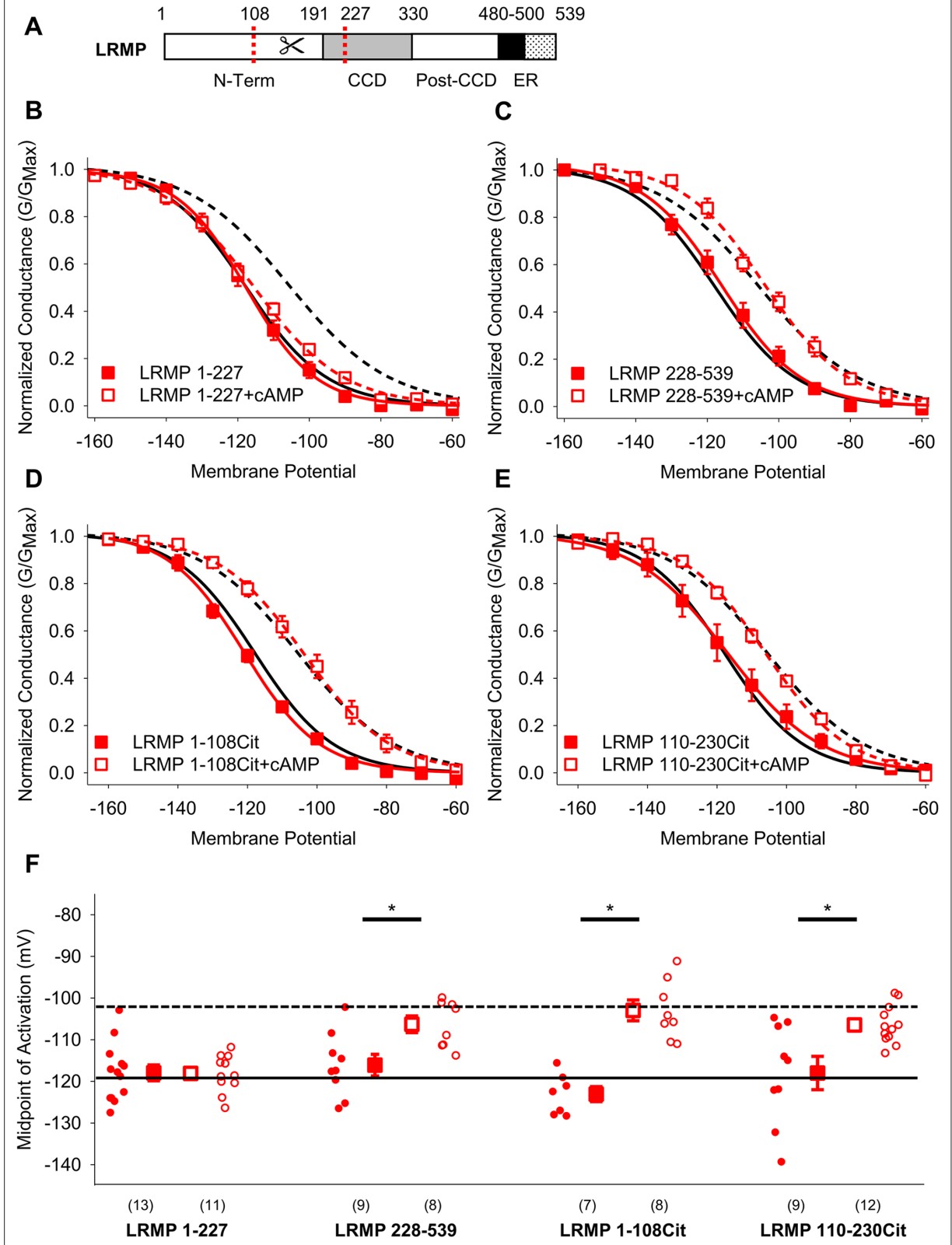

**Figure 2.** The pre-coiled-coil region of the LRMP N-terminus is necessary and sufficient to regulate HCN4. (**A**) Schematic of LRMP showing the coiled-coil domain (CCD) and ER-transmembrane and luminal domains (ER) as predicted by Alphafold (Q60664). The locations of cut sites in the LRMP coiled-coil and N-terminal domains are indicated (*red dotted lines*).( **B–E**) Voltage-dependence of activation for HCN4 in the absence (*black*) or presence (*red*) of LRMP 1–227 (**B**), LRMP 228–539 (**C**), LRMP 1-108Cit (**D**), or LRMP 110-230Cit (**E**), and/or 1 mM intracellular cAMP (*open symbols*). The midpoints of

*Figure 2 continued on next page*

*Figure 2 continued*

activation for HCN4 with (*dotted line*) or without (*solid line*) 1 mM cAMP in the absence of LRMP are shown. (**F**) Average (± standard error of the mean) midpoints of activation for HCN4 in the absence or presence of LRMP constructs and/or 1 mM cAMP using the same color scheme as (**B–E**). Small circles represent individual recordings and values in parentheses are the number of independent recordings for each condition. * indicates a significant (p<0.05) difference. All means, standard errors, and exact p-values are in *Table 1*.

The online version of this article includes the following figure supplement(s) for figure 2:

**Figure supplement 1.** Mouse LRMP sequence.

the lack of cAMP binding obviously prevented the investigation of any LRMP inhibition of cAMP-dependent potentiation (*Figure 4D and E*). While these results do not preclude a contribution of the C-terminus to modulation of HCN4 by LRMP, the persistent regulation when the distal C-terminus is truncated indicates that this region is not required.

## The N-terminus of LRMP associates with the N-terminus of HCN4

To test for physical association between different regions of LRMP and HCN4, we used a FRET-based hybridization assay called FRET two-hybrid that measures fluorescent energy transfer between fluorescent protein-tagged fragments of proteins expressed in cells. A similar approach has been used to define interactions between the N- and C-termini of EAG channels as well as between Calmodulin and $Ca_V1.2$ (*Gianulis et al., 2013*; *Erickson et al., 2003*). FRET two-hybrid has a number of advantages. First, detection of association between domains occurs in the native cellular environment. In addition, this approach decreases the false-negative rate by using short protein fragments to reduce distances between the fluorophores, thus reducing the potential for false negatives. Fragments of LRMP that were tagged on the C-terminus with Citrine were co-expressed in HEK293 cells with fragments of HCN4 tagged on the C-terminus with Cerulean (*Figure 5A*). We then measured FRET using the acceptor photobleaching method (*Bastiaens and Jovin, 1996*; *Wouters et al., 1998*; *Klipp et al., 2020*).

A Citrine-tagged construct corresponding to the functionally active domain of the LRMP N-terminus (LRMP NT, LRMP residues 1–230) did not significantly FRET with the full-length HCN4 N-terminus (NT, HCN4 residues 1–260; *Figure 5B*; *Table 3*). However, these fragments are large and likely unstructured, thus the fluorophores could be positioned at a distance greater than the range FRET can measure, which is ~20–80 Å. Indeed, when we expressed LRMP NT with halves of the HCN4 N-terminus — HCN4 N1 (residues 1–125) and HCN4 N2 (residues 126–260) — we measured significant FRET compared to control cells co-transfected with Cer-HCN4 fragments and Citrine alone (i.e., without any LRMP sequence; *Figure 5B*; *Table 3*). Halves of the LRMP N-terminus — LRMP L1 (residues 1–108) and LRMP L2 (residues 110–230) — also exhibited significant FRET with the whole HCN4 N-terminus and with HCN4 N-terminal fragments (*Figure 5C*; *Table 3*). No significant FRET was observed between LRMP fragments and a fragment of the HCN2 N-terminus that contains the conserved HCND and is analogous to HCN4 N2 (*Figure 5C*; *Table 3*). And none of the LRMP fragments tested exhibited significant FRET with an HCN4 C-Linker/CNBD construct compared to control experiments (*Figure 5B and C*; *Table 3*). Cerulean-tagged fragments of the distal C-terminus showed insufficient expression for FRET experiments. Ultimately these data suggest that the N-terminus of LRMP interacts with regions of the non-conserved distal N-terminus of HCN4.

## Mutants in the HCN4 C-linker disrupt LRMP's functional effects

Prior work has shown that transduction of cAMP-binding to shifts in channel activation require a tripartite interaction of a transduction centre comprised of the N-terminal HCND, the C-linker, and the S4-S5 linker (*Porro et al., 2019*; *Wang et al., 2020a*; *Kondapuram et al., 2022*). Since LRMP interacts with the HCN4 N-terminus and disrupts cAMP-dependent potentiation downstream of the cAMP binding site, we hypothesized that it may act via this cAMP transduction centre. Although the sequence of the transduction centre is highly conserved among HCN channel isoforms, the overall conformation of the region differs subtly between the known HCN channel structures of HCN1 and HCN4 (*Lee and MacKinnon, 2017*; *Saponaro et al., 2021*). We identified two HCN4-specific residues in the C-linker, P545 and T547, that could contribute to the HCN4-specific conformation, signal transduction, and LRMP regulation (*Figure 6A*). Mutation of these two residues to the cognate HCN2 amino acids, rendered the HCN4-P545A/T547F channel completely insensitive to LRMP, although it

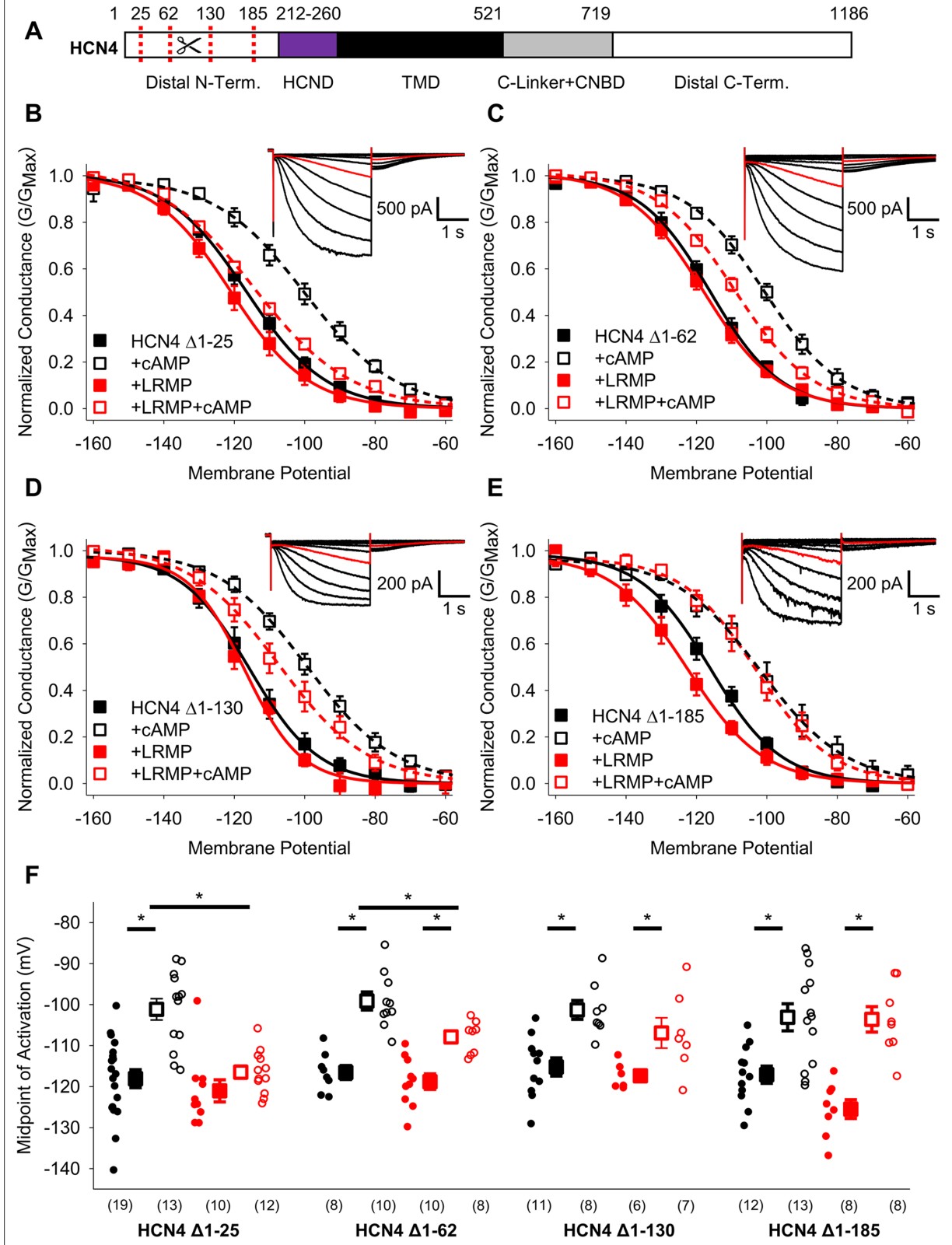

**Figure 3.** The distal HCN4 N-terminus is required for functional regulation by LRMP. (**A**) Schematic representation of HCN4 showing truncation sites (*red dotted lines*) in the non-conserved distal N-terminus (TMD: Transmembrane domain). (**B–E**) Voltage-dependence of activation for HCN4 Δ1–25 (**B**), HCN4 Δ1–62 (**C**), HCN4 Δ1–130 (**D**), and HCN4 Δ1–185 (**E**) in the absence (*black*) or presence of LRMP (*red*) and/or 1 mM intracellular cAMP (*open symbols*). (**B-E**) *Insets*: Exemplar current recordings for HCN4 Δ1–25 (**B**), HCN4 Δ1–62 (**C**), HCN4 Δ1–130 (**D**), and HCN4 Δ1–185 (**E**) in the absence of

*Figure 3 continued on next page*

*Figure 3 continued*

LRMP and cAMP. Currents recorded with a –110 mV activating pulse are shown in *red*. (**F**) Average (± standard error of the mean) midpoints of activation for HCN4 Δ1–25, HCN4 Δ1–62, HCN4 Δ1–130, and HCN4 Δ1–185 in the absence or presence of LRMP and/or 1 mM cAMP using the same color scheme as (**B–E**). Small circles represent individual recordings and values in parentheses are the number of independent recordings for each condition. * indicates a significant (p<0.05) difference. All means, standard errors, and exact p-values are in *Table 2*.

responded to cAMP with an ~10 mV shift in activation voltage (*Figure 6B and C*; *Table 2*), indicating that the unique residues in the HCN4 C-linker are important for LRMP regulation. However, the impact of these two HCN4 residues appears to depend on the overall context of the C-linker and C-terminus. A chimera containing the HCN4 N-terminus and transmembrane domains (residues 1–518) with the HCN2 C-linker, CNBD and C-terminus (442-863), termed HCN4-2 (*Liao et al., 2012*) was still partially regulated by LRMP (*Figure 6D and E*; *Table 2*). This result is satisfyingly consistent with our finding that deletion of the distal C-terminus in the HCN4-S719x construct reduces the overall effect of LRMP (*Figure 4B*). Together these data suggest that unique residues in the HCN4 C-linker are important for regulation by LRMP but that the distal C-terminus may also contribute to the regulation, potentially via allosteric effects on the orientation of the transduction centre.

## The HCN4 N-Terminus and cAMP transduction centre confer LRMP regulation to HCN2

Given that LRMP regulation involves HCN4-specific sequences in the C-linker region and distal N-terminus, we next investigated whether we could confer LRMP sensitivity to HCN2 by manipulating these regions. Mutation of the two non-conserved residues in the C-linker (HCN2 A467P/F469T) alone were not sufficient to confer regulation by LRMP onto HCN2 (*Figure 7A and B*; *Table 2*). In addition to the changes in the C-linker, comparison of the HCN1 and HCN4 structures shows different orientations of the S4-S5 region between HCN1 and HCN4 that may be responsible for differences in regulation of cAMP-sensitivity between channel isoforms (*Saponaro et al., 2021*). The S4-S5 linkers are fully conserved across HCN isoforms; however, three residues near the intracellular side of S5 differ between HCN2 and HCN4 (*Figure 6A*). We generated an HCN2 construct with all five non-conserved S5 and C-linker residues mutated to the corresponding HCN4 amino acids (HCN2 M338V/C341V/S345G/A467P/F469T). These mutations did not confer LRMP regulation to HCN2 (*Figure 7C and D*; *Table 2*), consistent with our data showing that the HCN4 N-terminus is required for LRMP regulation of channel gating (*Figure 3*) and may confer partial sensitivity to LRMP in HCN2 (*Figure 6*).

Finally, we made a chimeric HCN2 channel that contains the distal HCN4 N-terminus (residues 1–212, prior to the HCN domain) and the 5 non-conserved residues of the HCN4 S5 segment and C-linker elbow. The resulting HCN2-4N VVGPT channel has a voltage-dependence of activation similar to that of HCN2 and a normal response to cAMP in the absence of LRMP (*Figure 7E and F*; *Table 2*). However, the HCN2-4N VVGPT channel was fully regulated by LRMP — it became insensitive to cAMP in the presence of LRMP (*Figure 7E and F*; *Table 2*). Thus, the HCN4 N-terminus and a small number of HCN4-specific residues near the cAMP-transduction centre residues are sufficient to confer LRMP regulation to HCN2.

## Discussion

Ion channels families, such as HCN channels, are conveniently described by their shared properties. However, differences between isoforms underlie nuanced physiological functions of ion channels in tissues and present the opportunity for the design of drugs with higher specificity. Our present results reveal how subtle — and seemingly inconsequential — differences between isoforms in the same channel family can confer important differences in regulation. In the specific case of LRMP regulation of HCN4 channels, our data identify unique features of HCN4 that render its cAMP sensitivity particularly malleable, and thus could contribute to its unique function in the sinoatrial node of the heart.

Although LRMP prevents the cAMP-dependent shift in HCN4 activation, LRMP does not act by preventing cAMP from binding to the channel. Instead, we show here that the N-terminal domains of LRMP and HCN4 are required for both physical interaction and regulation. Our data further show that LRMP acts by disrupting transduction between cAMP binding and the shift in voltage-dependence

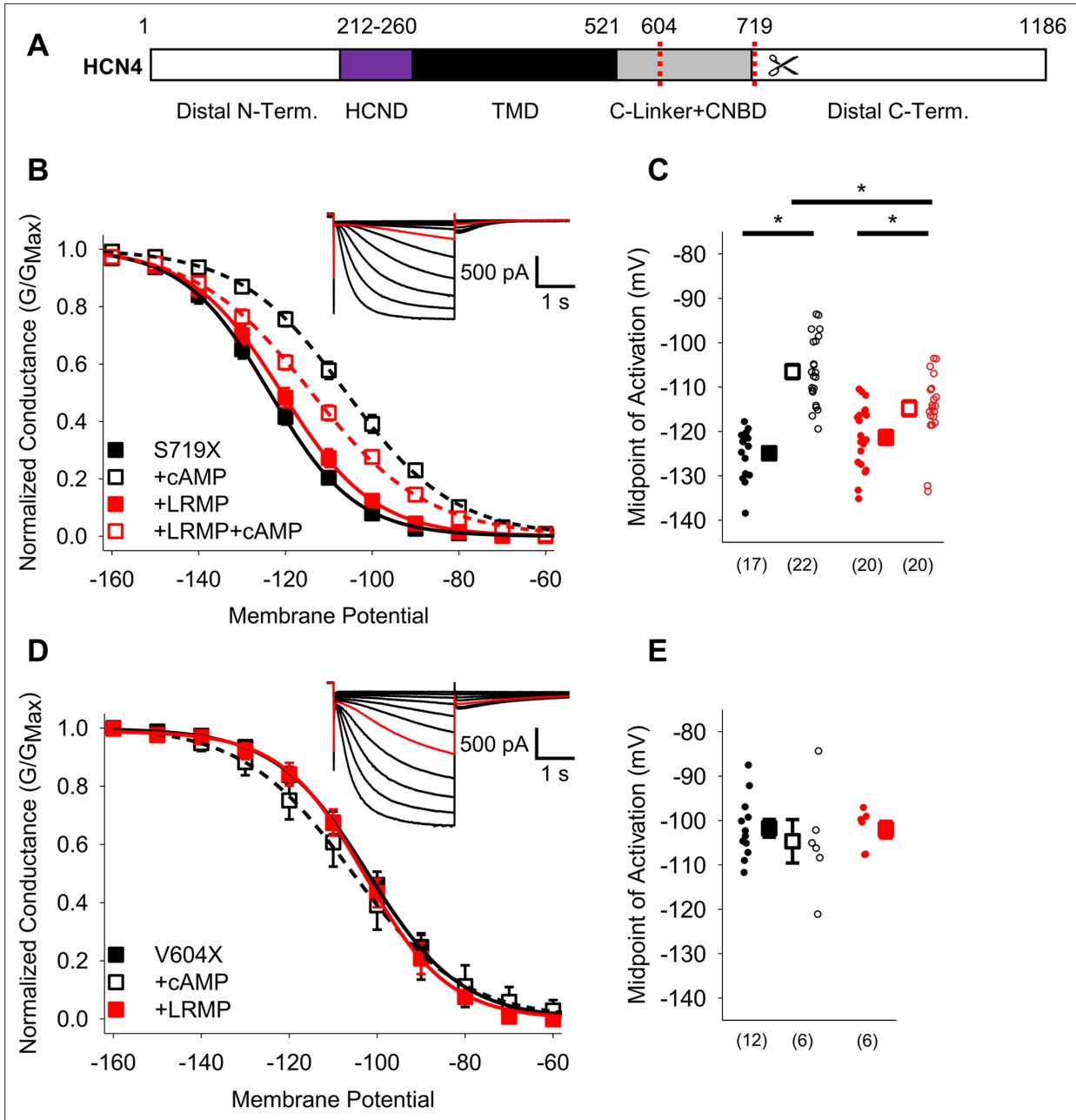

**Figure 4.** The HCN4 C-terminus is not the primary site for functional regulation by LRMP. (**A**) Schematic representation of HCN4 showing truncation sites (*red dotted lines*) of the distal C-terminus and CNBD (TMD: Transmembrane domain). (**B**) Voltage-dependence of activation for HCN4 S719X in the absence (*black*) or presence of LRMP (*red*) and/or 1 mM intracellular cAMP (*open symbols*). (**C**) Average (± standard error of the mean) midpoints of activation for HCN4 S719X in the absence or presence of LRMP and/or 1 mM cAMP using the same color scheme as (**B**). (**D**) Voltage-dependence of activation for HCN4 V604X in the absence or presence of LRMP or 1 mM intracellular cAMP using the same color scheme as (**B**). (**E**) Average (± standard error of the mean) midpoints of activation for HCN4 V604X in the absence or presence of LRMP or 1 mM cAMP using the same color scheme as (**B**). Small circles represent individual recordings in (**C**) and (**E**) and values in parentheses are the number of independent recordings for each condition. (**B and D**) *insets*: Exemplar current recordings for HCN4 S719X (**B**) and HCN4 V604X (**D**) in the absence of LRMP and cAMP. Currents recorded with a –110 mV activating pulse are shown in *red*. * indicates a significant (p<0.05) difference. All means, standard errors, and exact p-values are in *Table 2*.

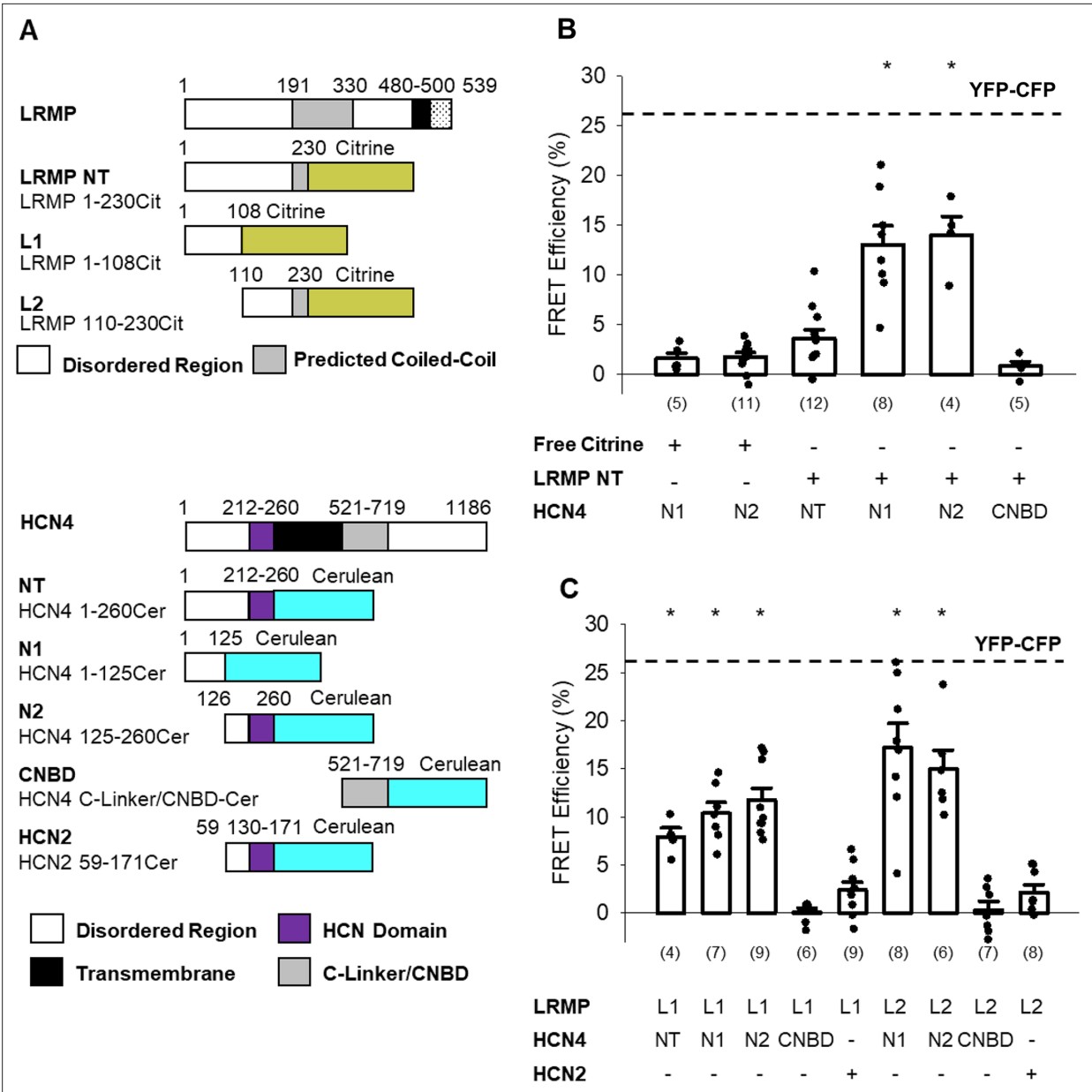

**Figure 5.** The N-terminus of LRMP FRETs with the N-terminus of HCN4. (**A**) Schematic representations of the Citrine-tagged LRMP fragments and Cerulean-tagged HCN4 and HCN2 fragments used in FRET experiments. (**B**) Average (± standard error of the mean) acceptor photobleaching FRET efficiency between free Citrine or the Citrine-tagged N-terminal region of the LRMP (LRMP NT) and the Cerulean-tagged HCN4 N-terminus (NT), halves of the HCN4 N-terminus (N1 and N2), or the HCN4 C-Linker/CNBD. The dotted line is the average FRET in YFP-CFP concatemers from a prior study (***Wang et al., 2020b***). (**C**) Average (± standard error of the mean) acceptor photobleaching FRET efficiency between Citrine-tagged fragments of the LRMP N-terminus (L1 and L2) and Cerulean-tagged fragments of HCN4 or HCN2. Small circles in (**B** and **C**) represent individual recordings and values in parentheses are the number of independent recordings for each condition. * indicates a significant (p<0.05) difference compared to control FRET in cells co-transfected with free Citrine and with Cerulean-tagged HCN4 N-terminal fragments. All means, standard errors, and exact p-values are in ***Table 3***.

**Table 3.** Acceptor photobleaching FRET between LRMP and HCN channel fragments.

| Citrine | Cerulean | FRET efficiency (%) | p-Value vs. control |
|---|---|---|---|
| Free Citrine | HCN4 1–125 or 125–260 | 1.7±0.3 | |
| | HCN4 1–125 | 1.6±0.6 (5) | |
| | HCN4 125–260 | 1.8±0.4 (11) | |
| LRMP 1–230 | HCN4 1–260 | 3.6±0.9 (12) | p=0.8471 |
| | HCN4 1–125 | 13.1±1.9 (8) | **p<0.0001** |
| | HCN4 125–260 | 14.0±1.9 (4) | **p<0.0001** |
| | HCN4 C-Linker/CNBD | 0.8±0.5 (5) | p=1.0000 |
| LRMP 1–108 | HCN4 1–260 | 7.9±1.0 (4) | **p=0.0265** |
| | HCN4 1–125 | 10.4±1.1 (7) | **p<0.0001** |
| | HCN4 125–260 | 11.7±1.3 (9) | **p<0.0001** |
| | HCN4 C-Linker/CNBD | 0.0±0.5 (6) | p=0.9869 |
| | HCN2 N-Term | 2.4±0.8 (9) | p=1.0000 |
| LRMP 110–230 | HCN4 1–125 | 17.2±2.5 (8) | **p<0.0001** |
| | HCN4 125–260 | 15.0±2.0 (6) | **p<0.0001** |
| | HCN4 C-Linker/CNBD | 0.3±0.9 (7) | p=0.9949 |
| | HCN2 N-Term | 2.1±0.8 (8) | p=1.0000 |

Average midpoint of activation ± standard error of the mean (Number of independent cells).

in a manner that depends on HCN4-specific residues in multiple domains, including the C-linker, S5, and C-terminus.

## An intramolecular transduction centre between the C-Linker, HCND, and S4-S5 linker links cAMP binding to shifts in activation

Recent studies indicate that binding of cAMP to the C-terminal CNBD is transduced to a shift in HCN channel activation via an intramolecular cAMP transduction centre formed by interactions between the C-linker, N-terminal HCND, and S4-S5 linker (*Weißgraeber et al., 2017*; *Porro et al., 2019*; *Wang et al., 2020a*; *Saponaro et al., 2021*; *Kondapuram et al., 2022*). There are multiple individual interactions within the transduction centre which have been described in detail (*Porro et al., 2019*; *Kondapuram et al., 2022*; *Elbahnsi et al., 2023*; *Wang et al., 2020a*). While most of these interactions are conserved among HCN channel isoforms, there are some isoform-specific differences that likely contribute to the unique sensitivity of HCN4 to LRMP and to other regulators that act to modify the cAMP response. Most notably, the S4-S5 linker of HCN4 adopts a different conformation compared to HCN1. Since the residues in the S4-S5 linker are completely conserved across HCN channel isoforms, we speculate that subtle differences in the sequence of nearby areas of the C-linker and S5 of HCN4 underlie the unique S4-S5 conformation of HCN4.

Despite often being grouped as the two cAMP-sensitive HCN isoforms, it is clear that cAMP signal transduction in HCN4 and HCN2 differs in several regards. The cAMP-dependent shift in HCN4 (~14 mV) is smaller than in HCN2 (~20 mV; *Table 2*). Transduction of cAMP binding is sensitive to divalent cations in HCN4 but not HCN2 (*Saponaro et al., 2021*; *Peters et al., 2023*). And, the cAMP-dependent shift in activation can be disrupted in HCN4 by regulatory factors such as cyclic-dinucleotides (*Lolicato et al., 2014*) as well as LRMP and IRAG (*Peters et al., 2020*). While there is not yet a structure available for HCN2, it is possible that differences in the orientation of the S4-S5 linker and the transduction centre makes HCN4 channels more sensitive to these perturbations in its cAMP signal transduction compared to HCN2.

## Proposed model: LRMP disrupts cAMP regulation of HCN4 activation at the cAMP transduction centre

In our proposed model for how LRMP disrupts the cAMP-dependent shift in HCN4 activation, LRMP is tethered to HCN4 via an interaction between the N-terminals of the two proteins. Within HCN4, the interaction with LRMP occurs via the distal N-terminus, which is not resolved in channel structures (*Lee and MacKinnon, 2017*; *Saponaro et al., 2021*) and is completely divergent between HCN channel isoforms. In contrast, the conserved HCND does not appear to participate in binding of LRMP; however, it is known to be an important component of the cAMP transduction centre (*Porro*

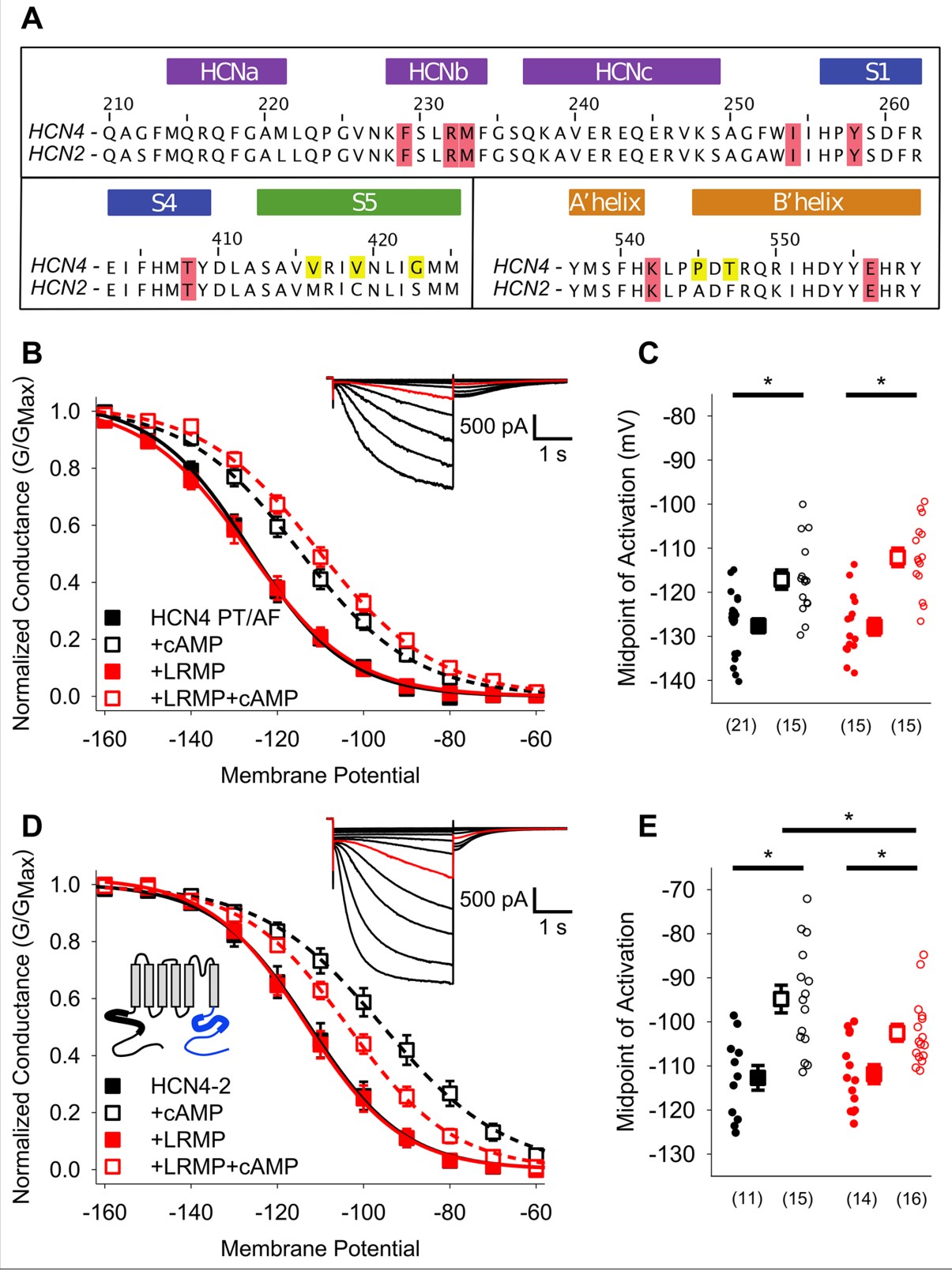

**Figure 6.** Mutants in the HCN4 C-linker disrupt LRMP's functional effects. (**A**) Sequence alignments of the HCN channel HCND (*purple*), voltage-sensor (*blue*), pore (*green*), and C-linker regions (*orange*) known to regulate cAMP-transduction. Non-conserved HCN4 residues in the S5 and C-linker regions are highlighted in yellow, and some of the residues believed to participate in cAMP-transduction are highlighted in red. (**B**) Voltage-dependence of activation for HCN4 P545A/T547F (PT/AF) in the absence (*black*) or presence of LRMP (*red*) and/or 1 mM intracellular cAMP (*open symbols*). (**C**) Average

*Figure 6 continued on next page*

*Figure 6 continued*

(± standard error of the mean) midpoints of activation for HCN4 PT/AF in the absence or presence of LRMP and/or 1 mM cAMP using the same color scheme as (**B**). (**D**) Voltage-dependence of activation for HCN4-2 (HCN4 1–518+HCN2 442-863) in the absence or presence of LRMP and/or 1 mM intracellular cAMP using the same color scheme as (**B**). *Schematic Inset*: Schematic of the chimeric HCN4-2 channel with HCN4 sequence shown in black and HCN2 in blue. The HCN and cyclic-nucleotide binding domains are indicated as thicker line segments. (**E**) Average (± standard error of the mean) midpoints of activation for HCN4-2 in the absence or presence of LRMP and/or 1 mM cAMP using the same color scheme as (**B**). (**B and D**) *insets*: Exemplar current recordings for HCN PT/AF (**B**) and HCN4-2 (**D**) in the absence of LRMP and cAMP. Currents recorded with a –110 mV activating pulse are shown in *red*. Small circles represent individual recordings in (**C**) and (**E**) and values in parentheses are the number of independent recordings for each condition. * indicates a significant (p<0.05) difference. All means, standard errors, and exact p-values are in *Table 2*.

*et al., 2019*; *Kondapuram et al., 2022*). It is also worth noting that the N-terminus of HCN4 is 260 amino acids long, compared to 140 and 209 in HCN1 and HCN2. Given the evolutionary and metabolic costs of maintaining long unique domains in these highly conserved proteins, it is possible that they serve other isoform-specific regulatory roles that await future discovery.

We found that truncation of the HCN4 N-terminus abolishes regulation by LRMP without affecting cAMP-dependent regulation. This finding was further corroborated by FRET experiments showing interactions between LRMP and the HCN4 N-terminus, but not the highly conserved CNBD. While these results do not preclude a modest role of the distal HCN4 C-terminus in LRMP regulation — as suggested by the partial LRMP regulation of the HCN4-2 and HCN4-719X channels — they clearly indicate that the N-terminus is critical for LRMP regulation of HCN4. Furthermore, the distal HCN4 C-terminus was not required to confer LRMP sensitivity to HCN2 (*Figure 7E and F*), suggesting that HCN4-specific residues in this region are not responsible for isoform-specific regulation by LRMP.

Most significantly, we were able to confer LRMP sensitivity to HCN2 by introducing only the HCN4 distal N-terminus and mutating five residues in the C-linker and S5 regions to the cognate HCN4 residues. It is highly unlikely that LRMP directly contacts the residues in S5, and our FRET experiments did not reveal an interaction with the C-linker/CNBD either (*Figure 5*). Thus, our data support a model in which LRMP interacts with the N-terminus of the HCN4 and prevents cAMP regulation of the channel allosterically, via effects on the unique transduction centre. We propose that the isoform specificity arises both from the unique distal N-terminal interaction site and from the unique orientation of the transduction centre in HCN4.

## Potential physiological implications

The first half of LRMP's cytosolic domain (residues 1–230) that make up the N-terminus of the protein is necessary and sufficient to interact with and regulate HCN4. Because the C-terminus of LRMP is embedded in the ER, the N-terminal region of LRMP would naturally be in closer proximity to HCN4 in the plasma membrane. LRMP also interacts with and regulates $Ca^{2+}$ release through inositol triphosphate ($IP_3$) receptors in the ER membrane, likely via a site in the coiled-coil region (*Prüschenk et al., 2021*). Together these results suggest the intriguing possibility of coordination between the activity of $IP_3Rs$ and HCN4 and the formation of ER-plasma membrane junctions in cells where LRMP and HCN4 are co-expressed. For example, in sinoatrial myocytes HCN4 and SR $Ca^{2+}$ release, including through $IP_3$ receptors, are both known to regulate pacemaking (*DiFrancesco, 2010*; *Peters et al., 2021*; *Capel et al., 2021*). A potential interaction with LRMP (or IRAG1), could serve to coordinate these important processes.

## Limitations

In this study, a FRET hybridization approach was used to identify macro-regions of LRMP and HCN4 that can interact with each other in a cellular context. It is important to acknowledge that this technique cannot resolve the atomic details of the interaction, which would ideally be addressed in the future by a co-structure of the proteins or at least their interaction domains. Other limitations of the approach are that the FRET efficiency measurement depends on the relative expression of each fragment, the affinity of the interaction, the orientation of the fluorophores, and the distance between the two fluorophores. This may explain why longer LRMP (LRMP 1-230Cit) and HCN4 (HCN4 1-260Cer) fragments showed lower FRET efficiency than did smaller fragments within these domains (*Figure 5*). Also, the approach may miss interactions that involve complex tertiary structures where binding

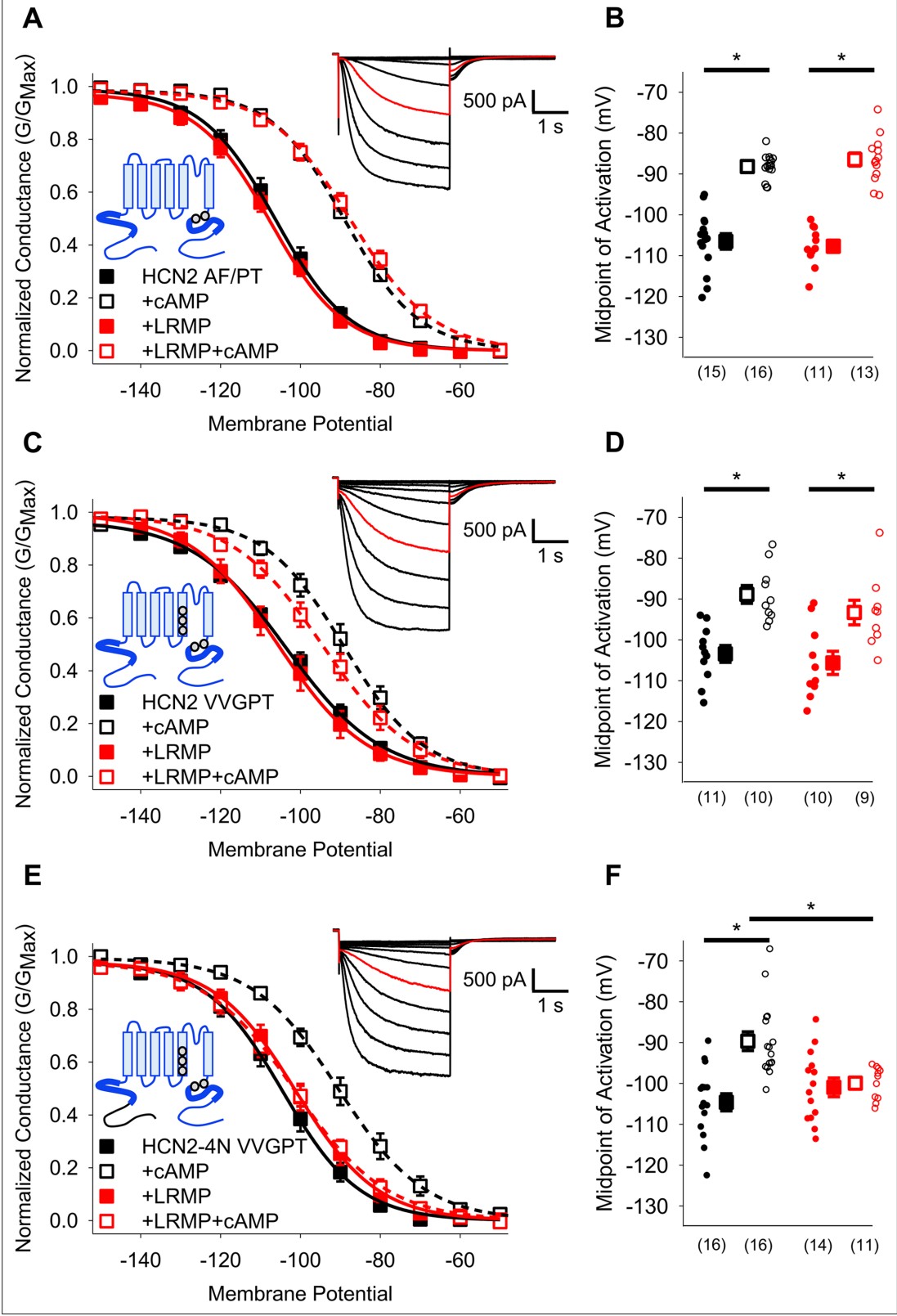

**Figure 7.** HCN4-specific residues and the HCN4 N-terminus confer LRMP regulation on HCN2. (**A**) Voltage-dependence of activation for HCN2 A467P/F469T (AF/PT) in the absence (*black*) or presence of LRMP (*red*) and/or 1 mM intracellular cAMP (*open symbols*). (**B**) Average (± standard error of the mean) midpoints of activation for HCN2 AF/PT in the absence or presence of LRMP and/or 1 mM cAMP using the same color scheme as (**A**). (**C**) Voltage-dependence of activation for HCN2 VVGPT (M338V/C341V/S345G/A467P/F469T) in the absence or presence of LRMP and/or 1 mM intracellular cAMP

*Figure 7 continued on next page*

*Figure 7 continued*

using the same color scheme as (**A**). (**D**) Average (± standard error of the mean) midpoints of activation for HCN2 VVGPT in the absence or presence of LRMP and/or 1 mM cAMP using the same color scheme as (**A**). (**E**) Voltage-dependence of activation for HCN2-4N VVGPT (HCN4 1–212+HCN2 135-863 M338V/C341V/S345G/A467P/F469T) in the absence or presence of LRMP and/or 1 mM intracellular cAMP using the same color scheme as (**A**). (**F**) Average (± standard error of the mean) midpoints of activation for HCN2-4N VVGPT in the absence or presence of LRMP and/or 1 mM cAMP using the same color scheme as (**A**). *Sample current insets*: Exemplar current recordings for HCN2 AF/PT (**A**), HCN2 VVGPT (**C**), and HCN2-4N VVGPT (**E**) in the absence of LRMP and cAMP. Currents recorded with a –110 mV activating pulse are shown in *red*. *Schematic Insets*: Schematics of the chimeric channels with HCN4 sequence shown in black and HCN2 in blue. The HCN and cyclic-nucleotide binding domains are indicated as thicker line segments. Small circles represent individual recordings in (**B, D**) and (**F**) and values in parentheses are the number of independent recordings for each condition. * indicates a significant (p<0.05) difference. All means, standard errors, and exact p-values are in *Table 2*.

involves multiple regions of the protein. Despite these limitations, our FRET and functional results together are consistent with the model that the N-terminals of HCN4 and LRMP directly interact.

Unfortunately, the available HCN4 structures do not resolve the distal N-terminus (*Shintre et al., 2018*; *Saponaro et al., 2021*), and the structure of LRMP has yet to be resolved. This lack of structural information hindered our decisions about specific cut sites for LRMP and HCN4 constructs and restricts our ability to predict the precise residues that are involved in the described interactions. For example, we found that LRMP interacts with isolated fragments representing each half of the HCN4 N-terminus. This could be explained by a diffuse interaction composed of multiple contacts, or our cut site overlapping a contiguous interaction site. The partial disruption of LRMP regulation of the HCN4 Δ1–62 and Δ1–130 deletion constructs suggests that multiple or diffuse interactions are likely. Similarly, we found that LRMP residues 1–108 and 110–230 both interacted with the HCN4 N-terminus in FRET assays, but neither fragment alone was able to regulate the channel. As with the HCN4 N-terminus, this difference could be explained by multiple important regions and/or our cut site overlapping the functionally relevant site. Ultimately, these questions will require a structure of the LRMP-HCN4 interaction interfaces.

## Summary

Overall, these data support a model for LRMP regulation of HCN4 where LRMP interacts with the HCN4 N-terminus to allosterically disrupt cAMP signal transduction between the C-linker, N-terminus, and S4-S5 linker (*Porro et al., 2019*; *Wang et al., 2020a*). The specific regulation of only the HCN4 isoform by LRMP is determined by both the non-conserved distal N-terminus and non-conserved residues in the C-linker and S5 of HCN4, which may result in a unique orientation of the cAMP transduction centre in HCN4. While a potential physiological role for LRMP regulation of HCN4 remains unknown, our data show that LRMP is a useful biophysical tool to study the intramolecular signal transduction between cAMP binding and the shift in HCN4 activation.

## Materials and methods

**Key resources table**

| Reagent type (species) or resource | Designation | Source or reference | Identifiers | Additional information |
|---|---|---|---|---|
| Cell line (*Homo-sapiens*) | HEK-293 | ATCC | CRL-1573 | |
| Cell line (*Homo-sapiens*) | HEK-HCN4 | Dr. Martin Biel; *Zong et al., 2012* | | |
| Cell line (*Homo-sapiens*) | HEK-HCN4 | This paper | ATCC CRL-1573; pcDNA3.1 mHCN4 | HEK-293 stably expressing HCN4 |
| Cell line (*Homo-sapiens*) | HEK-HCN2 | This paper | ATCC CRL-1573; pcDNA3.1 mHCN2 | HEK-293 stably expressing HCN2 |
| Cell line (*Homo-sapiens*) | HEK-HCN4 Δ1–62 | This paper | ATCC CRL-1573; pTwist-CMV-WPRE-Neo mHCN4 Δ1–62 | HEK-293 stably expressing HCN4 Δ1–62 |
| Cell line (*Homo-sapiens*) | HEK-HCN4 Δ1–130 | This paper | ATCC CRL-1573; pTwist-CMV-WPRE-Neo mHCN4 Δ1–130 | HEK-293 stably expressing HCN4 Δ1–130 |

*Continued on next page*

*Continued*

| Reagent type (species) or resource | Designation | Source or reference | Identifiers | Additional information |
|---|---|---|---|---|
| Cell line (*Homo-sapiens*) | HEK-HCN4 Δ1–185 | This paper | ATCC CRL-1573; pTwist-CMV-WPRE-Neo mHCN4 Δ1–185 | HEK-293 stably expressing HCN4 Δ1–185 |
| Cell line (*Homo-sapiens*) | HEK-HCN4 Δ1–200 | This paper | ATCC CRL-1573; pTwist-CMV-Hygro mHCN4 Δ1–200 | HEK-293 stably expressing HCN4 Δ1–200 |
| Cell line (*Homo-sapiens*) | HEK-HCN4 PT/AF | This paper | ATCC CRL-1573; pcDNA3.1 mHCN4 PT/AF | HEK-293 stably expressing HCN4 P545A/T547F |
| Cell line (*Homo-sapiens*) | HEK-HCN2 AF/PT | This paper | ATCC CRL-1573; pTwist-CMV-WPRE-Neo mHCN2 AF/PT | HEK-293 stably expressing HCN2 A467P/F469T |
| Recombinant DNA reagent | pcDNA3.1 mHCN1 | Dr. Eric Accili; *Proenza et al., 2002* | | |
| Recombinant DNA reagent | pcDNA3.1 mHCN1 | This paper; *Liao et al., 2012* | | mHCN2 (sequence NP_032252.1) subcloned from pcDNA4 |
| Recombinant DNA reagent | pcDNA6 mHCN4 Δ1–25 | Dr. Richard Aldrich; *Liu and Aldrich, 2011* | | |
| Recombinant DNA reagent | pTwist-CMV-WPRE-Neo mHCN2 A467P/F469T | This paper | | Synthesized by Twist Bioscience based on sequence NP_032252.1 |
| Recombinant DNA reagent | pTwist-CMV-WPRE-Neo mHCN4 | This paper | NP_001074661.1; codon optimized | Synthesized by Twist Bioscience |
| Recombinant DNA reagent | pTwist-CMV-WPRE-Neo mHCN4 Δ1–62 | This paper | | Deletions made using site-directed mutagenesis in pTwist-CMV-WPRE-Neo HCN4 |
| Recombinant DNA reagent | pTwist-CMV-WPRE-Neo mHCN4 Δ1–130 | This paper | | Deletions made using site-directed mutagenesis in pTwist-CMV-WPRE-Neo HCN4 |
| Recombinant DNA reagent | pTwist-CMV-WPRE-Neo mHCN4 Δ1–185 | This paper | | Deletions made using site-directed mutagenesis in pTwist-CMV-WPRE-Neo HCN4 |
| Recombinant DNA reagent | pTwist-CMV-Hygro mHCN4 Δ1–200 | This paper | | Synthesized by Twist Bioscience based on sequence NP_001074661.1 |
| Recombinant DNA reagent | pcDNA3.1 mHCN4 P545T/A547F | This paper | | Site-directed mutagenesis of pcDNA3.1 HCN4 by Applied Biological Materials |
| Recombinant DNA reagent | pcDNA3.1 mHCN4 V604X | This paper | | Site-directed mutagenesis of pcDNA3.1 HCN4 by Applied Biological Materials |
| Recombinant DNA reagent | pcDNA3.1 mHCN4 S719X | Proenza Lab; *Liao et al., 2012* | | |
| Recombinant DNA reagent | pcDNA4 mHCN4-2 | Proenza Lab; *Liao et al., 2012* | | HCN4 residues 1–518 plus HCN2 residues 442–863 |
| Recombinant DNA reagent | pTwist-CMV-BG-WPRE-Neo mHCN2-4N VVGPT | This paper | | Synthesized by Twist Bioscience based on sequence NP_001074661.1 and NP_032252.1 |
| Recombinant DNA reagent | pTwist-CMV-BG-WPRE-Neo mHCN2 VVGPT | This paper | | Synthesized by Twist Bioscience based on sequence NP_032252.1 |
| Recombinant DNA reagent | pCMV6 Kan/Neo mLRMP | Origene | CAT#: MC201923 | Untagged mouse LRMP construct |
| Recombinant DNA reagent | pCMV6 Kan/Neo Myc-mLRMP | Proenza Lab; *Peters et al., 2020* | | N-terminal Myc-tagged LRMP construct |
| Recombinant DNA reagent | pTwist-CMV mLRMP 1–227 | This paper | | Synthesized by Twist Bioscience based on GenBank AAH52909.1 |

*Continued on next page*

*Continued*

| Reagent type (species) or resource | Designation | Source or reference | Identifiers | Additional information |
|---|---|---|---|---|
| Recombinant DNA reagent | pTwist-CMV mLRMP 228–539 | This paper | | Synthesized by Twist Bioscience based on GenBank AAH52909.1 |
| Recombinant DNA reagent | pcDNA3.1 mHCN4 125-260Cer | This paper | | C-terminal Cerulean; see DNA constructs section of the methods |
| Recombinant DNA reagent | pcDNA3.1 mHCN4 1-260Cer | This paper | | C-terminal Cerulean; see DNA constructs section of the methods |
| Recombinant DNA reagent | pcDNA3.1 mHCN4 521-719Cer | This paper | | C-terminal Cerulean; see DNA constructs section of the methods |
| Recombinant DNA reagent | pcDNA3.1 mHCN4 1-125Cer | This paper | | C-terminal Cerulean; see DNA constructs section of the methods |
| Recombinant DNA reagent | pcMVBG mLRMP 1-479Cit | This paper | | C-terminal Citrine; see DNA constructs section of the methods |
| Recombinant DNA reagent | pcMVBG mLRMP 1-230Cit | This paper | | C-terminal Citrine; see DNA constructs section of the methods |
| Recombinant DNA reagent | pcMVBG mLRMP 1-108Cit | This paper | | C-terminal Citrine; see DNA constructs section of the methods |
| Recombinant DNA reagent | pcMVBG mLRMP 110-230Cit | This paper | | C-terminal Citrine; see DNA constructs section of the methods |
| Commercial assay or kit | Q5 Site-Directed Mutagenesis Kit | New England Biolabs | CAT#: E0554S | |
| Commercial assay or kit | In-Fusion HD Cloning | Clontech | Clontech:639647 | |
| Chemical compound, drug | FuGENE 6 | Promega | CAT#: E2691 | |
| Chemical compound, drug | Lipofectamine 2000 | Thermo-Fisher Scientific | CAT#: 11668027 | |
| Software, Algorithm | pClamp and clampfit | Molecular Devices | RRID:SCR_011323 | |
| Software, Algorithm | ImageJ | NIH DOI: https://doi.org/10.1038/nmeth.2089 | RRID:SCR_003070 | |
| Software, Algorithm | Sigmaplot 12.0 | Systat Software Inc | RRID:SCR_003210 | |
| Software, Algorithm | JMP14 | SAS Institute | RRID:SCR_014242 | |

## DNA constructs

The mouse LRMP construct in PCMV6-Kan/Neo (GenBank AAH52909.1; Cat. #MC228229, Origene, Rockville, MD; *Figure 2—figure supplement 1*), HCN1 in pcDNA3 (generously provided by Dr. Eric Accili), HCN4-2 in pcDNA3.1, HCN4-S719X in pCDNA3.1, and HCN4 Δ1–25 in pcDNA6 (also known as HCN4s, generously provided by Dr. Richard Aldrich) have been described previously (*Proenza et al., 2002*; *Liao et al., 2012*; *Liu and Aldrich, 2011*; *Peters et al., 2020*). HCN2 was subcloned from pcDNA4 into pcDNA3.1 for this study. Other constructs were synthesized by Twist Biosciences (South San Francisco, CA) or using site-directed mutagenesis either in-house or by Applied Biological Materials (Richmond, Canada). The HCN4 Δ1–62, HCN4 Δ1–130, and HCN4 Δ1–185 deletion clones were made using a site-directed mutagenesis kit (New England Biolabs, Ipswich, MA) and a codon-optimized HCN4 plasmid in the pTwist-CMV-WPRE-Neo vector synthesized by Twist Biosciences.

For FRET experiments, recombinant fusions of mHCN4 and mLRMP were constructed by introducing Cerulean (CER) or Citrine (CIT) fluorescent proteins using PCR-based cloning. The C-termini of HCN4 constructs were tagged with CER, while the C-termini of LRMP constructs were tagged with CIT. Because there are no structures of LRMP or the or the N-terminus of HCN4, and because much of the experimental design was carried out prior to Alphafold's structural predictions (*Jumper et al., 2021*; *Varadi et al., 2022*), the specific cut sites were determined empirically. We tried a number of

fragments and the ones used in this study were of similar sizes and expressed well in our system. The sites we chose relative to the predicted coiled-coil on LRMP can be seen in *Figure 2—figure supplement 1*.

All clones used in this study are the murine sequences of LRMP and HCN4. All new constructs were confirmed by DNA sequencing (Barbara Davis Center BioResource Core at the University of Colorado Anschutz Medical Campus; ACGT, Wheeling, IL; or Plasmidsaurus, Eugene, OR). Detailed information about constructs can be found in the key resources table.

## Cell lines

HEK293 cells were obtained from ATCC, which uses STR profiling for cell line authentication. HEK 293 cells from which new cell lines were established and HEK HCN4 cells were negative for mycoplasma infection. Testing for mycoplasma infection was performed at the Molecular Biology Core Facility in the Barbara Center for Childhood Diabetes at the University of Colorado Anschutz Medical Campus. None of the cells are on the list of commonly misidentified cell lines.

HEK 293 cells (ATCC, Manassas, VA) were grown in a humidified incubator at 37 °C and 5% $CO_2$ in high glucose DMEM with L-glutamine supplemented with 10% FBS, 100 U/mL penicillin, and 100 µg/mL streptomycin. Cells were transfected 48 hr prior to experiments and were plated on either protamine-coated glass coverslips (for patch clamp experiments) or poly-d-lysine coated glass-bottom dishes (for FRET experiments).

Patch clamp experiments were performed in either transiently transfected HEK293 cells, an HCN4 stable line in HEK293 cells (*Zong et al., 2012*), or eight new stable cell lines in HEK293 cells: HCN2, HCN4, HCN4 Δ1–62, HCN4 Δ1–130, HCN4 Δ1–185, HCN4 Δ1–200, HCN2 A467P/F469T (HCN2 AF/ PT), and HCN4 P545A/T547F (HCN4 PT/AF). Stable cell lines were made by transfecting HEK293 cells with the respective plasmids using Lipofectamine 2000 (Invitrogen, Waltham, MA) according to the manufacturer's instructions. Forty-eight hours post-transfection, 200 µg/mL of G418 disulfate (Alfa Aesar, Haverhill, MA) or Hygromycin B (InvivoGen, San Diego, CA) was added to the cell culture media in place of pen-strep to select for stably transfected cells. Single-cell clones were tested using whole-cell patch clamp and the clonal lines that exhibited the largest and most consistent currents were grown into stable cell lines. Control experiments of HCN4 in the absence of LRMP were conducted alongside recordings in the presence of LRMP to ensure that stably expressed channels had consistent properties over the time course of the study.

Transient transfection of HCN4 constructs and/or LRMP was performed using Fugene6 (Promega, Madison, WI) according to the manufacturer's instructions. Transfections of all constructs that did not include fluorescent tags were performed with the addition of eGFP (at a LRMP to GFP ratio of 4:1) as a co-transfection marker. All data were collected from a minimum of 3 transfections per condition. The N-values listed in *Tables 1–3* represent the number of individual cells that were patch-clamped for a given condition.

## Patch clamp electrophysiology

Cells were plated on sterile protamine-coated glass coverslips 24–48 hr prior to experiments. Cells on coverslip shards were transferred to the recording chamber and perfused (~0.5–1 mL/min) with extracellular solution containing (in mM): 30 KCl, 115 NaCl, 1 $MgCl_2$, 1.8 $CaCl_2$, 5.5 glucose, and 5 HEPES. Transiently transfected cells were identified by green fluorescence.

Patch pipettes were pulled from borosilicate glass to a resistance of 1.0–3.0 MOhm when filled with intracellular solution containing (in mM): 130 K-Aspartate, 10 NaCl, 1 EGTA, 0.5 $MgCl_2$, 5 HEPES, and 2 Mg-ATP. One mM cAMP was added to the intracellular solution as indicated. All recordings were performed at room temperature in the whole-cell configuration. Data were acquired at 5 KHz, and low-pass filtered at 10 KHz using an Axopatch 200B amplifier (Molecular Devices, San Jose, CA), Digidata 1440 A A/D converter and Clampex software (Molecular Devices). Pipette capacitance was compensated in all recordings. Membrane capacitance and series resistance (Rs) were estimated in whole-cell experiments using 5 mV test pulses. Only cells with a stable Rs of <10 MOhm were analyzed. Data were analyzed in Clampfit 10.7 (Molecular Devices).

Channel activation was determined from peak tail current amplitudes at –50 mV following 3 s hyperpolarizing pulses to membrane potentials between –50 mV and –170 mV from a holding potential of 0 mV. Normalized tail current-voltage relationships were fit by single Boltzmann equations to

yield values for the midpoint activation voltage ($V_{1/2}$) and slope factor (k). Deactivation time constants were determined using a single exponential fit to tail currents recorded at –50 mV following a pulse to –150 mV. All reported voltages are corrected for a calculated +14 mV liquid junction potential between the extracellular and intracellular solutions.

## FRET hybridization assays

HEK293 cells expressing HCN4-CER and LRMP-CIT fusion proteins were examined 48 hr after transfection using a Zeiss LSM 710 confocal laser scanning microscope. An area of 500–2,500 μm$^2$ was selected from the overall field of view. Images were taken through a 40×water objective. CER and CIT were excited with separate sweeps of the 458- and 514 nm laser lines of an argon laser directed at the cell with a 458/514 nm dual dichroic mirror. Relative to full power, the excitation power for the imaging sweeps was attenuated to 1% for CER and 0.5% for Citrine. Bleaching was performed by using multiple (20–60) sweeps of the CIT laser at full power. Bleaching was usually complete within 1–2 m. Emitted light was collected between 449 and 488 nm for CER and 525 and 600 nm for CIT. With this setup, there was no contamination of the relevant CER signal from the CIT. For each experiment, the photomultiplier tube gain was adjusted to ensure that the maximum pixel intensity was not >70% saturated. Fluorescence intensity was then measured by drawing regions of interest (ROIs) around the cytoplasmic portion of the cell in ImageJ (*Schneider et al., 2012*). Masks were occasionally used to eliminate bright fluorescent puncta within the cell (this was a rare occurrence in the CER signal). Percent FRET (E) was calculated as:

$$E = \left[ I_{CER_{post}} - I_{CER_{pre}} \right] / I_{CER_{post}} 100,$$

where $I_{CER_{post}}$ is the CER intensity after bleaching and $I_{CER_{pre}}$ is the CER intensity before bleaching.

## Statistical analysis

All statistical analysis was performed using JMP 15 software (SAS Institute, Cary, NC). Normality was tested using the Shapiro-Wilk test. The log of the deactivation time constant at –50 mV was used for statistical analysis to ensure the data were normally distributed. To prevent biasing of the results, all data were included except for cells showing large changes in leak or access resistance during the recording, or those for which the access resistance was >10 MΩ at any point during recording. Tests for differences in the average midpoint of activation for a given HCN channel construct in the presence of LRMP and/or 1 mM cAMP were performed with a 2-way ANOVA. The main independent variables were the absence or presence of LRMP and the absence or presence of 1 mM cAMP in the pipette solution. Differences in the effects of cAMP in the absence or presence of LRMP were analyzed using an interaction term between the main independent variables. For FRET experiments, the recordings of Cerulean tagged HCN4 1–125 and HCN4 125–260 co-expressed with free Citrine were pooled as a control group. $p < 0.05$ was used as the cut-off for a significant effect. All comparisons meeting this criteria are indicated in figures by an asterisks, with exact p-values given in the manuscript text or *Tables 1–3*.

## Materials Availability

All new cell lines and plasmids used in this study are described in the key resources table and are stored in the Bankston and Proenza laboratories at the University of Colorado Anschutz Medical Campus. Cell lines and plasmids can be accessed by contacting either of the corresponding authors.

## Acknowledgements

This work was funded by NIH grants R01HL088427 and R01GM140004 to CP and R35GM137912 to JB. CHP was funded by an American Heart Association Postdoctoral Fellowships 830889 and 19POST34380777. The authors gratefully acknowledge the contributions of Abby Camenisch and Karin Nunley.

## Additional information

### Funding

| Funder | Grant reference number | Author |
|---|---|---|
| National Institute of General Medical Sciences | R35GM137912 | John R Bankston |
| National Institute of General Medical Sciences | R01GM140004 | Catherine Proenza |
| National Heart, Lung, and Blood Institute | R01HL088427 | Catherine Proenza |
| American Heart Association | 830889 | Colin H Peters |
| American Heart Association | 19POST34380777 | Colin H Peters |

The funders had no role in study design, data collection and interpretation, or the decision to submit the work for publication.

### Author contributions

Colin H Peters, Conceptualization, Data curation, Formal analysis, Supervision, Funding acquisition, Investigation, Writing – original draft, Writing – review and editing; Rohit K Singh, Conceptualization, Data curation, Formal analysis, Investigation; Avery A Langley, William G Nichols, Hannah R Ferris, Danielle A Jeffrey, Data curation, Formal analysis, Investigation; Catherine Proenza, Conceptualization, Data curation, Supervision, Funding acquisition, Investigation, Writing – original draft, Project administration, Writing – review and editing; John R Bankston, Conceptualization, Data curation, Formal analysis, Supervision, Funding acquisition, Writing – original draft, Project administration, Writing – review and editing

### Author ORCIDs

Colin H Peters ⓘ https://orcid.org/0000-0001-8557-9100
Hannah R Ferris ⓘ http://orcid.org/0000-0002-5675-5336
Catherine Proenza ⓘ https://orcid.org/0000-0003-4324-6206
John R Bankston ⓘ http://orcid.org/0000-0002-9478-2335

Reviewer #1 (Public Review): https://doi.org/10.7554/eLife.92411.3.sa1
Reviewer #2 (Public Review): https://doi.org/10.7554/eLife.92411.3.sa2
Reviewer #3 (Public Review): https://doi.org/10.7554/eLife.92411.3.sa3
Author response https://doi.org/10.7554/eLife.92411.3.sa4

## Additional files

### Supplementary files

• Source data 1. Individual data points, averages, and standard errors of the mean for patch-clamp and FRET data.

• MDAR checklist

### Data availability

All data generated or analysed during this study are included in the manuscript and supporting files; source data files have been provided as well.

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
