## [Editor Report · eLife assessment]

This study identifies the molecular determinants of LRMP co-regulation of HCN 4 activity. The evidence supporting the conclusions, which is **compelling**, is backed by rigorous electrophysiological and spectroscopic analysis. The work is **important** because it greatly enhances our understanding of the mechanisms of HCN channel regulation in a tissue-specific manner and highlights a functional role for more disordered regions that have yet to be structurally resolved.

---

## [Referee Report · Reviewer #1 (Public Review)]

Summary:

The authors use truncations, fragments, and HCN2/4 chimeras to narrow down the interaction and regulatory domains for LRMP inhibition of cAMP-dependent shifts in the voltage dependence of activation of HCN4 channels. They identify the N-terminal domain of HCN4 as a binding domain for LRMP, and highlight two residues in the C-linker as critical for the regulatory effect. Notably, whereas HCN2 is normally insensitive to LRMP, putting the N-terminus and 5 additional C-linker and S5 residues from HCN4 into HCN2 confers LRMP regulation in HCN2.

Strengths:

The work is excellent, the paper well written, and the data convincingly support the conclusions which shed new light on the interaction and mechanism for LRMP regulation of HCN4, as well as identifying critical differences that explain why LRMP does not regulate other isoforms such as HCN2.

---

## [Referee Report · Reviewer #2 (Public Review)]

Summary:

HCN-4 isoform is found primarily in sino-atrial node where it contributes to the pacemaking activity. LRMP is an accessory subunit which prevents cAMP-dependent potentiation of HCN4 isoform but does not have any effect on HCN2 regulation. In this study, the authors combine electrophysiology, FRET with standard molecular genetics to determine the molecular mechanism of LRMP action on HCN4 activity. Their study shows parts of N- and C-termini along with specific residues in C-linker and S5 of HCN4 are crucial for mediating LRMP action on these channels. Furthermore, they show that the initial 224 residues of LRMP are sufficient to account for most of the activity. In my view, the highlight of this study is Fig. 7 which recapitulates LRMP modulation on HCN2-HCN4 chimera. Overall, this study is an excellent example of using time-tested methods to probe the molecular mechanisms of regulation of channel function by an accessory subunit.

The authors adequately addressed my earlier concerns.

---

## [Referee Report · Reviewer #3 (Public Review)]

Summary:

Using patch clamp electrophysiology and Förster resonance energy transfer (FRET), Peters and co-workers showed that the disordered N-terminus of both LRMP and HCN4 are necessary for LRMP to interact with HCN4 and inhibit the cAMP-dependent potentiation of channel opening. Strikingly, they identified two HCN4-specific residues, P545 and T547 in the C-linker of HCN4, that are close in proximity to the cAMP transduction centre (elbow Clinker, S4/S5-linker, HCND) and account for the LRMP effect.

Strengths:

Based on these data, the Authors propose a mechanism in which LRMP specifically binds to HCN4 via its isotype-specific Nterminal sequence and thus prevents the cAMP transduction mechanism by acting at the interface between the elbow Clinker, the S4S5-linker, the HCND.

Weaknesses:

Although the work is interesting, there are some discrepancies between data that need to be addressed.

- I suggest inserting in Table 1 and in the text, the Δ shift values (+cAMP; + LRMP; +cAMP/LRMP). This will help readers.

- Figure 1 is not clear, the distribution of values is anomalously high. For instance, in 1B the distribution of values of V1/2 in the presence of cAMP goes from - 85 to -115. I agree that in the absence of cAMP, HCN4 in HEK293 cells shows some variability in V1/2 values, that nonetheless cannot be so wide (here the variability spans sometimes even 30 mV) and usually disappears with cAMP (here not).

This problem is spread throughout the ms, and the measured mean effects indeed always at the limit of statistical significance. Why so? Is this a problem with the analysis, or with the recordings?

There are several other problems with Figure 1 and in all figures of the ms: the Y scale is very narrow while the mean values are marked with large square boxes. Moreover, the exemplary activation curve of Fig 1A is not representative of the mean values reported in Figure 1B, and the values of 1B are different from those reported in Table 1.

On this ground it is difficult to judge the conclusions and it would also greatly help if exemplary current traces would also be shown.

- "....HCN4-P545A/T547F was insensitive to LRMP (Figs. 6B and 6C; Table 1), indicating that the unique HCN4 C-linker is necessary for regulation by LRMP. Thus, LRMP appears to regulate HCN4 by altering the interactions between the C-linker, S4-S5 linker, and N-terminus at the cAMP transduction centre."

Although this is an interesting theory, there are no data supporting it. Indeed, P545 and T547 at the tip of the C-linker elbow (fig 6A) are crucial for LRMP effect, but these two residues are not involved in the cAMP transduction centre (interface between HCND, S4S5 linker and Clinker elbow), at least for the data accumulated till now in the literature. Indeed, the hypothesis that LRMP somehow inhibits the cAMP transduction mechanism of HCN4 given the fact that the two necessary residues P545 and T547 are close to the cAMP transduction centre, awaits to be proven.

Moreover, I suggest analysing the putative role of P545 and T547 in the light of the available HCN4 structures. In particular, T547 (elbow) point towards the underlying shoulder of the adjacent subunit and, therefore, it is in a key position for the cAMP transduction mechanism. The presence of bulky hydrophobic residues (very different nature compared to T) in the equivalent position of HCN1 and HCN2 is also favouring this hypothesis. In this light, it will also be interesting to see whether single T547F mutation is sufficient to prevent LRMP effect.

---

## [Author Response]

The following is the authors’ response to the original reviews.

**eLife assessment**
This is a useful study examining the determinants and mechanisms of LRMP inhibi:on of cAMP regula:on of HCN4 channel ga:ng. The evidence provided to support the main conclusions is unfortunately incomplete, with discrepancies in the work that reduce the strength of mechanis:c insights.

Thank you for the reviews of our manuscript. We have made a number of changes to clarify our hypotheses in the manuscript and addressed all of the poten:al discrepancies by revising some of our interpreta:on. In addi:on, we have provided addi:onal experimental evidence to support our conclusions. Please see below for a detailed response to each reviewer comment.

**Public Reviews**

**Reviewer #1 (Public Review):**
Summary:The authors use truncations, fragments, and HCN2/4 chimeras to narrow down the interaction and regulatory domains for LRMP inhibition of cAMP-dependent shifts in the voltage dependence of activation of HCN4 channels. They identify the N-terminal domain of HCN4 as a binding domain for LRMP, and highlight two residues in the C-linker as critical for the regulatory effect. Notably, whereas HCN2 is normally insensitive to LRMP, putting the N-terminus and 5 additional C-linker and S5 residues from HCN4 into HCN2 confers LRMP regulation in HCN2.Strengths:The work is excellent, the paper well written, and the data convincingly support the conclusions which shed new light on the interaction and mechanism for LRMP regulation of HCN4, as well as identifying critical differences that explain why LRMP does not regulate other isoforms such as HCN2.

Thank you.

**Reviewer #2 (Public Review):**
Summary:HCN-4 isoform is found primarily in the sino-atrial node where it contributes to the pacemaking activity. LRMP is an accessory subunit that prevents cAMP-dependent potentiation of HCN4 isoform but does not have any effect on HCN2 regulation. In this study, the authors combine electrophysiology, FRET with standard molecular genetics to determine the molecular mechanism of LRMP action on HCN4 activity. Their study shows that parts of N- and C-termini along with specific residues in C-linker and S5 of HCN4 are crucial for mediating LRMP action on these channels. Furthermore, they show that the initial 224 residues of LRMP are sufficient to account for most of the activity. In my view, the highlight of this study is Fig. 7 which recapitulates LRMP modulation on HCN2-HCN4 chimera. Overall, this study is an excellent example of using time-tested methods to probe the molecular mechanisms of regulation of channel function by an accessory subunit.Weaknesses:(1) Figure 5A- I am a bit confused with this figure and perhaps it needs better labeling. When it states Citrine, does it mean just free Citrine, and "LRMP 1-230" means LRMP fused to Citrine which is an "LF" construct? Why not simply call it "LF"? If there is no Citrine fused to "LRMP 1-230", this figure would not make sense to me.

We have clarified the labelling of this figure and specifically defined all abbreviations used for HCN4 and LRMP fragments in the results section on page 14.

(2) Related to the above point- Why is there very little FRET between NF and LRMP 1-230? The FRET distance range is 2-8 nm which is quite large. To observe baseline FRET for this construct more explanation is required. Even if one assumes that about 100 amino are completely disordered (not extended) polymers, I think you would still expect significant FRET.

FRET is extremely sensitive to distance (to the 6th power of distance). The difference in contour length (maximum length of a peptide if extended) between our ~260aa fragment and our ~130 aa fragments is on the order of 450Å (45nm), So, even if not extended it is not hard to imagine that the larger fragments show a weaker FRET signal. In fact, we do see a slightly larger FRET than we do in control (not significant) which is consistent with the idea that the larger fragments just do not result in a large FRET.

Moreover, this hybridization assay is sensitive to a number of other factors including the affinity between the two fragments, the expression of each fragment, and the orientation of the fluorophores. Any of these factors could also result in reduced FRET.

We have added a section on the limitations of the FRET 2-hybrid assay in the discussion section on page 20. Our goal with the FRET assay was to provide complimentary evidence that shows some of the regions that are important for direct association and we have edited to the text to make sure we are not over-interpreting our results.

(3) Unless I missed this, have all the Cerulean and Citrine constructs been tested for functional activity?

All citrine-tagged LRMP constructs (or close derivatives) were tested functionally by coexpression with HCN (See Table 1 and pages 10-11). Cerulean-tagged HCN4 fragments are of course intrinsically not-functional as they do not include the ion conducting pore.

**Reviewer #3 (Public Review):**
Summary:Using patch clamp electrophysiology and Förster resonance energy transfer (FRET), Peters and co-workers showed that the disordered N-terminus of both LRMP and HCN4 are necessary for LRMP to interact with HCN4 and inhibit the cAMP-dependent potentiation of channel opening. Strikingly, they identified two HCN4-specific residues, P545 and T547 in the C-linker of HCN4, that are close in proximity to the cAMP transduction centre (elbow Clinker, S4/S5-linker, HCND) and account for the LRMP effect.Strengths:Based on these data, the authors propose a mechanism in which LRMP specifically binds to HCN4 via its isotype-specific N-terminal sequence and thus prevents the cAMP transduction mechanism by acting at the interface between the elbow Clinker, the S4S5-linker, the HCND.Weaknesses:Although the work is interesting, there are some discrepancies between data that need to be addressed.(1) I suggest inserting in Table 1 and in the text, the Δ shift values (+cAMP; + LRMP; +cAMP/LRMP). This will help readers.

Thank you, Δ shift values have been added to Tables 1 and 2 as suggested.

(2) Figure 1 is not clear, the distribution of values is anomalously high. For instance, in 1B the distribution of values of V1/2 in the presence of cAMP goes from - 85 to -115. I agree that in the absence of cAMP, HCN4 in HEK293 cells shows some variability in V1/2 values, that nonetheless cannot be so wide (here the variability spans sometimes even 30 mV) and usually disappears with cAMP (here not).

With a large N, this is an expected distribution. In 5 previous reports from 4 different groups of HCN4 with cAMP in HEK 293 (Fenske et al., 2020; Liao et al., 2012; Peters et al., 2020; Saponaro et al., 2021; Schweizer et al., 2010), the average expected range of the data is 26.6 mV and 39.9 mV for 95% (mean ± 2*SD) and 99% (mean ± 3*SD) of the data, respectively. As the reviewer mentions the expected range from these papers is slightly larger in the absence of cAMP. The average SD of HCN4 (with/without cAMP) in papers are 9.9 mV (Schweizer et al., 2010), 4.4 mV (Saponaro et al., 2021), 7.6 mV (Fenske et al., 2020), 10.0 mV (Liao et al., 2012), and 5.9 mV (Peters et al., 2020). Our SD in this paper is roughly in the middle at 7.6 mV. This is likely because we used an inclusive approach to data so as not to bias our results (see the statistics section of the revised manuscript on page 9). We have removed 2 data points that meet the statistical classification as outliers, no measures of statistical significance were altered by this.

This problem is spread throughout the manuscript, and the measured mean effects are indeed always at the limit of statistical significance. Why so? Is this a problem with the analysis, or with the recordings?

The exact P-values are NOT typically at the limit of statistical significance, about 2/3rds would meet the stringent P < 0.0001 cut-off. We have clarified in the statistics section (page 10) that any comparison meeting our significance threshold (P < 0.05) or a stricter criterion is treated equally in the figure labelling. Exact P-values are provided in Tables 1-3.

There are several other problems with Figure 1 and in all figures of the manuscript: the Y scale is very narrow while the mean values are marked with large square boxes. Moreover, the exemplary activation curve of Figure 1A is not representative of the mean values reported in Figure 1B, and the values of 1B are different from those reported in Table 1.

Y-axis values for mean plots were picked such that all data points are included and are consistent across all figures. They have been expanded slightly (-75 to -145 mV for all HCN4 channels and -65 to -135 mV for all HCN2 channels). The size of the mean value marker has been reduced slightly. Exact midpoints for all data are also found in Tables 1-3.

The GV curves in Figure 1B (previously Fig. 1A) are averages with the ± SEM error bars smaller than the symbols in many cases owing to relatively high n’s for these datasets. These curves match the midpoints in panel 1C (previously 1B). Eg. the midpoint of the average curve for HCN4 control in panel A is -117.9 mV, the same as the -117.8 mV average for the individual fits in panel B.

We made an error in the text based on a previous manuscript version about the ordering of the tables that has now been fixed so these values should now be aligned.

On this ground, it is difficult to judge the conclusions and it would also greatly help if exemplary current traces would be also shown.

Exemplary current traces have been added to all figures in the revised manuscript.

(3) "....HCN4-P545A/T547F was insensitive to LRMP (Figs. 6B and 6C; Table 1), indicating that the unique HCN4 C-linker is necessary for regulation by LRMP. Thus, LRMP appears to regulate HCN4 by altering the interactions between the C-linker, S4-S5 linker, and Nterminus at the cAMP transduction centre."Although this is an interesting theory, there are no data supporting it. Indeed, P545 and T547 at the tip of the C-linker elbow (fig 6A) are crucial for LRMP effect, but these two residues are not involved in the cAMP transduction centre (interface between HCND, S4S5 linker, and Clinker elbow), at least for the data accumulated till now in the literature. Indeed, the hypothesis that LRMP somehow inhibits the cAMP transduction mechanism of HCN4 given the fact that the two necessary residues P545 and T547 are close to the cAMP transduction centre, remains to be proven.Moreover, I suggest analysing the putative role of P545 and T547 in light of the available HCN4 structures. In particular, T547 (elbow) points towards the underlying shoulder of the adjacent subunit and, therefore, is in a key position for the cAMP transduction mechanism. The presence of bulky hydrophobic residues (very different nature compared to T) in the equivalent position of HCN1 and HCN2 also favours this hypothesis. In this light, it will be also interesting to see whether a single T547F mutation is sufficient to prevent the LRMP effect.

We agree that testing this hypothesis would be very interesting. However, it is challenging. Any mutation we make that is involved in cAMP transduction makes measuring the LRMP effect on cAMP shifts difficult or impossible.

Our simple idea, now clarified in the discussion, is that if you look at the regions involved in cAMP transduction (HCND, C-linker, S4-S5), there are very few residues that differ between HCN4 and HCN2. When we mutate the 5 non-conserved residues in the S5 segment and the C-linker, along with the NT, we are able to render HCN2 sensitive to LRMP. Therefore, something about the small sequence differences in this region confer isoform specificity to LRMP. We speculate that this happens because of small structural differences that result from those 5 mutations. If you compare the solved structures of HCN1 and HCN4 (there is no HCN2 structure available), you can see small differences in the distances between key interacting residues in the transduction centre. Also, there is a kink at the bottom of the S4 helix in HCN4 but not HCN1. This points a putatively important residue for cAMP dependence in a different direction in HCN4. We hypothesize in the discussion that this may be how LRMP is isoform specific.

Moreover, previous work has shown that the HCN4 C-linker is uniquely sensitive to di-cyclic nucleotides and magnesium ions. We are hypothesizing that it is the subtle change in structure that makes this region more prone to regulation in HCN4.

**Reviewing Editor (recommendations for the Authors):**
(1) Exemplar recordings need to be shown and some explanation for the wide variability in the V-half of activation.

Exemplar currents are now shown for each channel. See the response to Reviewer 3’s public comment 2.

(2) The rationale for cut sites in LRMP for the investigation of which parts of the protein are important for blocking the effect of cAMP is not logically presented in light of the modular schematics of domains in the protein (N-term, CCD, post-CCD, etc).

There is limited structural data on LRMP and the HCN4 N-terminus. The cut sites in this paper were determined empirically. We made fragments that were small enough to work for our FRET hybridization approach and that expressed well in our HEK cell system. The residue numbering of the LRMP modules is based on updated structural predictions using Alphafold, which was released after our fragments were designed. This has been clarified in the methods section on pages 5-6 and the Figure 2 legend of the revised manuscript.

(3) Role of the HCN4 C-terminus. Truncation of the HCN4 C-terminus unstructured Cterminus distal to the CNBD (Fig. 4 A, B) partially reverses the impact of LRMP (i.e. there is now a significant increase in cAMP effect compared to full-length HCN4). The manuscript is written in a manner that minimizes the potential role of the C-terminus and it is, therefore, eliminated from consideration in subsequent experiments (e.g. FRET) and the discussion. The model is incomplete without considering the impact of the C-terminus.

We thank the reviewer for this comment as it was a result that we too readily dismissed. We have added discussion around this point and revised our model to suggest that not only can we not eliminate a role for the distal C-terminus, our data is consistent with it having a modest role. Our HCN4-2 chimera and HCN4-S719x data both suggest the possibility that the distal C-terminus might be having some effect on LRMP regulation. We have clarified this in the results (pages 12-13) and discussion (page 19).

(4) For FRET experiments, it is not clear why LF should show an interaction with N2 (residues 125-160) but not NF (residues 1-160). N2 is contained within NF, and given that Citrine and Cerulean are present on the C-terminus of LF and N2/NF, respectively, residues 1-124 in NF should not impact the detection of FRET because of greater separation between the fluorophores as suggested by the authors.

This is a fair point but FRET is somewhat more complicated. We do not know the structure of these fragments and it’s hard to speculate where the fluorophores are oriented in this type of assay. Moreover, this hybridization assay is sensitive to affinity and expression as well. There are a number of reasons why the larger 1-260 fragment might show reduced FRET compared to 125-260. As mentioned in our response to reviewer 2’s public comment 2, we have added a limitation section that outlines the various caveats of FRET that could explain this.

(5) For FRET experiments, the choice of using pieces of the channel that do not correlate with the truncations studied in functional electrophysiological experiments limits the holistic interpretation of the data. Also, no explanation or discussion is provided for why LRMP fragments that are capable of binding to the HCN4 N-terminus as determined by FRET (e.g. residues 1-108 and 110-230, respectively) do not have a functional impact on the channel.

As mentioned in the response to comment 2, the exact fragment design is a function of which fragments expressed well in HEK cells. Importantly, because FRET experiments do not provide atomic resolution for the caveats listed in the revised limitations section on page 20-21, small differences in the cut sites do not change the interpretation of these results. For example, the N-terminal 1-125 construct is analogous to experiments with the Δ1-130 HCN4 channel.

We suspect that residues in both fragments are required and that the interaction involves multiple parts. This is stated in the results “Thus, the first 227 residues of LRMP are sufficient to regulate HCN4, with residues in both halves of the LRMP N-terminus necessary for the regulation” (page 11). We have also added discussion on this on page 21.

(6) A striking result was that mutating two residues in the C-linker of HCN4 to amino acids found in HCN channels not affected by LRMP (P545A, T547F), completely eliminated the impact of LRMP on preventing cAMP regulation of channel activation. However, a chimeric channel, (HCN4-2) in which the C-linker, the CNBD, and the C-terminus of HCN4 were replaced by that of HCN2 was found to be partially responsive to LRMP. These two results appear inconsistent and not reconciled in the model proposed by the authors for how LRMP may be working.

As stated in our answer to your question #3, we have revised our interpretation of these data. If the more distal C-terminus plays some role in the orientation of the C-linker and the transduction centre as a whole, these data can still be viewed consistent with our model. We have added some discussion of this idea in our discussion section.

(7) Replacing the HCN2 N-terminus with that from HCN4, along with mutations in the S5 (MCS/VVG) and C-linker (AF/PT) recapitulated LRMP regulation on the HCN2 background. The functional importance of the S5 mutations is not clear as no other experiments are shown to indicate whether they are necessary for the observed effect.

We have added our experiments on a midpoint HCN2 clone that includes the S5 mutants and the C-linker mutants in the absence of the HCN4 N-terminus (ie HCN2 MCSAF/VVGPT) (Fig. 7). And we have discussed our rationale for the S5 mutations as we believe they may be responsible for the different orientations of the S4-S5 linker in HCN1 and HCN4 structures that are known to impact cAMP regulation.

**Reviewer #1 (Recommendations For The Authors):**
A) Comments:(1) Figure 1: Please show some representative current traces.

Exemplar currents are now shown for each channel in the manuscript.

(2) Figure 1: There appears to be a huge number of recordings for HCN4 +/- cAMP as compared to those with LRMP 1-479Cit. How was the number of recordings needed for sufficient statistical power decided? This is particularly important because the observed slowing of deactivation by cAMP in Fig. 1C seems like it may be fairly subtle. Perhaps a swarm plot would make the shift more apparent? Also, LRMP 1-479Cit distributions in Fig. 1B-C look like they are more uniform than normal, so please double-check the appropriateness of the statistical test employed.

We have revised the methods section (page 7) to discuss this, briefly we performed regular control experiments throughout this project to ensure that a normal cAMP response was occurring. Our minimum target for sufficient power was 8-10 recordings. We have expanded the statistics section (page 9) to discuss tests of normality and the use of a log scale for deactivation time constants which is why the shifts in Fig. 1D (revised) are less apparent.

(3) It would be helpful if the authors could better introduce their logic for the M338V/C341V/S345G mutations in the HCN4-2 VVGPT mutant.

See response to the reviewing editor’s comment 7.

B) Minor Comments:(1) pg. 9: "We found that LRMP 1-479Cit inhibited HCN4 to an even greater degree than the full-length LRMP, likely because expression of this tagged construct was improved compared to the untagged full-length LRMP, which was detected by co-transfection with GFP." Co-transfection with GFP seems like an extremely poor and a risky measure for LRMP expression.

We agree that the exact efficiency of co-transfection is contentious although some papers and manufacturer protocols indicate high co-transfection efficiency (Xie et al., 2011). In this paper we used both co-transfection and tagged proteins with similar results.

(2) pg 9: "LRMP 1-227 construct contains the N-terminus of LRMP with a cut-site near the Nterminus of the predicted coiled-coil sequence". In Figure 2 the graphic shows the coiledcoil domain starting at 191. What was the logic for splitting at 227 which appears to be the middle of the coiled-coil?

See response to the reviewing editor’s comment 2.

(3) Figure 5C: Please align the various schematics for HCN4 as was done for LRMP. It makes it much easier to decipher what is what.

Fig. 5 has been revised as suggested.

(4) pg 12: I assume that the HCN2 fragment chosen aligns with the HCN4 N2 fragment which shows binding, but this logic should be stated if that is the case. If not, then how was the HCN2 fragment chosen?

This is correct. This has been explicitly stated in the revised manuscript (page 14).

(5) Figure 7: Add legend indicating black/gray = HCN4 and blue = HCN2.

This has been stated in the revised figure legend.

(6) pg 17: Conservation of P545 and T547 across mammalian species is not shown or cited.

This sentence is not included in the revised manuscript, however, for the interest of the reviewer we have provided an alignment of this region across species here.

**Author response image 1. sa4fig1:** 

**Reviewer #2 (Recommendations For The Authors):**
(1) It is not clear whether in the absence of cAMP, LRMP also modestly shifts the voltagedependent activity of the channels. Please clarify.

We have clarified that LRMP does not shift the voltage-dependence in the absence of cAMP(page 10). In the absence of cAMP, LRMP does not significantly shift the voltagedependence of activation in any of the channels we have tested in this paper (or in our prior 2020 paper).

(2) Resolution of Fig. 8b is low.

We ultimately decided that the cartoon did not provide any important information for understanding our model and it was removed.

(3) Please add a supplementary figure showing the amino acid sequence of LRMP to show where the demarcations are made for each fragment as well as where the truncations were made as noted in Fig 3 and Fig 4.

A new supplementary figure showing the LRMP sequence has been added and cited in the methods section (page 5). Truncation sites have been added to the schematic in Fig. 2A.

(4) In the cartoon schematic illustration for Fig. 3 and Fig.4, the legend should include that the thick bold lines in the C-Terminal domain represent the CNBD, while the thick bold lines in the N-Terminal domain represent the HCN domain. This was mentioned in Liao 2012, as you referenced when you defined the construct S719X, but it would be nice for the reader to know that the thick bold lines you have drawn in your cartoon indicate that it also highlights the CNBD or the HCN domain.

This has been added to figure legends for the relevant figures in the revised manuscript.

(5) On page 12, missing a space between "residues" and "1" in the parenthesis "...LRMP L1 (residues1-108)...".

Fixed. Thank you.

(6) Which isoform of LRMP was used? What is the NCBI accession number? Is it the same one from Peters 2020 ("MC228229")?

This information has been added to the methods (page 5). It is the same as Peters 2020.

**Reviewer #3 (Recommendations For The Authors):**
(1) "Truncation of residues 1-62 led to a partial LRMP effect where cAMP caused a significant depolarizing shift in the presence of LRMP, but the activation in the presence of LRMP and cAMP was hyperpolarized compared to cAMP alone (Fig. 3B, C and 3E; Table 1). In the HCN4Δ1-130 construct, cAMP caused a significant depolarizing shift in the presence of LRMP; however, the midpoint of activation in the presence of LRMP and cAMP showed a non-significant trend towards hyperpolarization compared to cAMP alone (Fig. 3C and 3E; Table 1)".This means that sequence 62-185 is necessary and sufficient for the LRMP effect. I suggest a competition assay with this peptide (synthetic, or co-expressed with HCN4 full-length and LRMP to see whether the peptide inhibits the LRMP effect).

We respectfully disagree with the reviewer’s interpretation. Our results, strongly suggest that other regions such as residues 25-65 (Fig. 3C) and C-terminal residues (Fig. 6) are also necessary. The use of a peptide could be an interesting future experiment, however, it would be very difficult to control relative expression of a co-expressed peptide. We think that our results in Fig. 7E-F where this fragment is added to HCN2 are a better controlled way of validating the importance of this region.

(2) "Truncation of the distal C-terminus (of HCN4) did not prevent LRMP regulation. In the presence of both LRMP and cAMP the activation of HCN4-S719X was still significantly hyperpolarized compared to the presence of cAMP alone (Figs. 4A and 4B; Table 1). And the cAMP-induced shift in HCN4-S719X in the presence of LRMP (~7mV) was less than half the shift in the absence of LRMP (~18 mV)."On the basis of the partial effects reported for the truncations of the N-terminus of HCN4 162 and 1-130 (Fig 3B and C), I do not think it is possible to conclude that "truncation of the distal C-terminus (of HCN4) did not prevent LRMP regulation". Indeed, cAMP-induced shift in HCN4 Δ1-62 and Δ1-130 in the presence of LRMP were 10.9 and 10.5 mV, respectively, way more than the ~7mV measured for the HCN4-S719X mutant.As you rightly stated at the end of the paragraph:" Together, these results show significant LRMP regulation of HCN4 even when the distal C-terminus is truncated, consistent with a minimal role for the C-terminus in the regulatory pathway". I would better discuss this minimal role of the C-terminus. It is true that deletion of the first 185 aa of HCN4 Nterminus abolishes the LRMP effect, but it is also true that removal of the very Cterm of HCN4 does affect LRMP. This unstructured C-terminal region of HCN4 contains isotype-specific sequences. Maybe they also play a role in recognizing LRMP. Thus, I would suggest further investigation via truncations, even internal deletions of HCN4-specific sequences.

Please see the response to the reviewing editor’s comment 3.

(3) Figure 5: The N-terminus of LRMP FRETs with the N-terminus of HCN4.Why didn't you test the same truncations used in Fig. 3? Indeed, based on Fig 3, sequences 1-25 can be removed. I would have considered peptides 26-62 and 63-130 and 131-185 and a fourth (26-185). This set of peptides will help you connect binding with the functional effects of the truncations tested in Fig 3.

Please see the response to the reviewing editor’s comment 2 and 5.

Why didn't you test the C-terminus (from 719 till the end) of HCN4? This can help with understanding why truncation of HCN4 Cterminus does affect LRMP, tough partially (Fig. 4A).

Please see the response to the reviewing editor’s comment 3.

(4) "We found that a previously described HCN4-2 chimera containing the HCN4 N-terminus and transmembrane domains (residues 1-518) with the HCN2 C-terminus (442-863) (Liao et al., 2012) was partially regulated by LRMP (Fig. 7A and 7B)".I do not understand this partial LRMP effect on the HCN4-2 chimera. In Fig. 6 you have shown that the "HCN4-P545A/T547F was insensitive to LRMP (Figs. 6B and 6C; Table 1), indicating that the unique HCN4 C-linker is necessary for regulation by LRMP". How can be this reconciled with the HCN4-2 chimera? HCN4-2, "containing" P545A/T547F mutations, should not perceive LRMP.

Please see the response to the reviewing editor’s comment 6.

(5) "we next made a targeted chimera of HCN2 that contains the distal HCN4 N-terminus (residues 1-212) and the HCN2 transmembrane and C-terminal domains with 5 point mutants in non-conserved residues of the S5 segment and C-linker elbow (M338V/C341V/S345G/A467P/F469T)......Importantly, the HCN4-2 VVGPT channel is insensitive to cAMP in the presence of LRMP (Fig. 7C and 7D), indicating that the HCN4 Nterminus and cAMP-transduction centre residues are sufficient to confer LRMP regulation to HCN2".Why did you insert also the 3 mutations of S5? Are these mutations somehow involved in the cAMP transduction mechanism?You have already shown that in HCN4 only P545 and T547 (Clinker) are necessary for LRMP effect. I suggest to try, at least, the chimera of HCN2 with only A467P/F469T. They should work without the 3 mutations in S5.

Please see the response to the reviewing editor’s comment 7.